# AliMark: Enhancing Robustness of Sentence-Level Watermarking Against Text Paraphrasing

**Yuexin Li** [* 1]  **Wenjie Qu** [* 1]  **Linyu Wu** [1]  **Yulin Chen** [1]  **Yufei He** [1]  **Tri Cao** [1]  **Bryan Hooi** [1]  **Jiaheng Zhang** [1]

## Abstract

Existing sentence-level watermarking methods enhance robustness to paraphrasing by anchoring watermarks in sentence semantics. However, their prefix-based designs remain vulnerable to structural perturbations, such as sentence splitting and merging, which commonly arise under strong paraphrasers like DIPPER and GPT-3.5. To mitigate this issue, we propose *AliMark*, a framework that reformulates sentence-level watermarking as a bit sequence encoding and alignment problem between a potentially watermarked text and a secret bit sequence. Notably, our approach adopts a two-stage detection strategy: we generate multiple restructured text variants and adaptively align their extracted bit sequences with the secret bit sequence to minimize alignment cost. This multi-candidate alignment design naturally improves robustness to sentence merges and splits. Extensive experiments demonstrate that AliMark substantially outperforms state-of-the-art baselines under diverse paraphrasing attacks. Our code is available at https://github.com/imethanlee/AliMark.

## 1. Introduction

The rapid advancement of Large Language Models (LLMs) has enabled high-quality text generation at scale (Yang et al., 2025; Guo et al., 2025; Singh et al., 2025). Alongside these benefits, the inability to distinguish machine-generated content poses significant risks to intellectual property and academic integrity (Mitchell et al., 2023). Consequently, LLM watermarking has emerged as a vital mechanism for copyright tracing and plagiarism prevention, offering a technical solution for content provenance and enforcing accountability in an era of automated writing (Kirchenbauer et al., 2023).

Current watermarking strategies primarily focus on the token level, embedding imperceptible signals by manipulating token sampling probabilities during decoding, such as KGW and its variants (Kirchenbauer et al., 2023; 2024). Despite their efficiency, these token-level watermarks suffer from a critical vulnerability: they are inherently fragile against adversarial rewriting. Malicious actors can easily remove the watermark using paraphrasing models (Krishna et al., 2023; Damodaran, 2021; Zhang et al., 2020) that replace specific tokens with synonyms, destroying the statistical biases required for detection. To address this fragility, sentence-level watermarking shifts the embedding target to the semantic space (Hou et al., 2024a;b; Ren et al., 2024; Dabiriaghdam & Wang, 2025). By anchoring the watermark in the underlying meaning rather than surface lexical choices, these approaches ensure the signal persists despite paraphrasing, offering a robust defense where token-based methods fail.

**Challenges**. Existing sentence-level watermarking methods, such as SemStamp (Hou et al., 2024a), inherit KGW's prefix-based design, in which each sentence's watermark signal is pseudorandomly conditioned on its preceding context. However, as shown in Section 2.2, advanced paraphrasers (OpenAI, 2022; Krishna et al., 2023) frequently induce *structural perturbations*, such as sentence splitting or merging, thereby altering the semantic contexts of the other unaffected sentences significantly. When the context of an unaffected sentence changes, it destroys the pseudorandom relationships previously held between the sentence and its original context; hence, detecting a watermark for that unaffected sentence becomes highly unlikely. Likewise, detecting a watermark for that changed context is also improbable. Consequently, a single context change can potentially cascade into the loss of watermark signals across multiple sentences, rendering existing frameworks inherently fragile under structural perturbations.

**Present Work**. To mitigate this issue, we propose AliMark, a sentence-level watermarking framework designed for enhanced resilience against structural perturbations induced by paraphrasing. Our approach draws inspiration from (Kuditipudi et al., 2024), which mitigates the impact of token insertion and deletion in KGW-style watermarking

---

[*]Equal contribution  [1]National University of Singapore. Correspondence to: Yuexin Li <yuexinli@u.nus.edu>.

*Proceedings of the 43rd International Conference on Machine Learning*, Seoul, South Korea. PMLR 306, 2026. Copyright 2026 by the author(s).

by reformulating detection, moving from simple green–red token counting to a sequence alignment problem. This formulation effectively limits the impact of changing context tokens on the tokens themselves, thereby improving watermarking robustness. Building on this insight, we recast sentence-level watermarking as bit sequence encoding and alignment tasks, achieving analogous robustness to changes in sentence context.

Specifically, AliMark embeds watermark signals into a sentence by selecting sentences whose bit signals, obtained via a bit signals extractor, correspond to a designated block of a secret bit sequence. This process ensures that the entire watermarked text encodes a sequence of bit signals. During watermark detection, to account for potential structural perturbations, the input text is first processed by a Re-Structurer, which generates multiple text variants through re-merging and re-splitting attempts to restore the original text structure. Each variant is then analyzed by an Adaptive Bit Sequence Alignment module, which aligns its extracted bit sequence with the secret bit sequence of varying lengths. The alignment cost is defined using Block Edit Rate, an adaptation of Levenshtein Distance (Levenshtein et al., 1966) for sentence-level scenarios. These text-level manipulations and bit-level alignment strategies aim to minimize the alignment cost with the secret bit sequence, thereby mitigating the effects of structural perturbations introduced by paraphrasing attacks on the original watermarked text.

We conduct extensive experiments to validate the effectiveness of AliMark over multiple watermarking baselines. Notably, AliMark outperforms baselines under most paraphrasing attacks, and the performance gain becomes much more significant under stronger paraphrasers like DIPPER (Krishna et al., 2023) and GPT-3.5 (OpenAI, 2022).

In summary, we make the following contributions: (1) We show that structural perturbations, such as sentence splitting and merging, are common in automated paraphrasing and can catastrophically break existing sentence-level watermarking methods. (2) We propose *AliMark*, a new formulation of the sentence-level watermarking task as bit sequence encoding and alignment, to mitigate the vulnerability to structural perturbations. (3) Extensive experiments validate that AliMark outperforms state-of-the-art baselines under multiple paraphrasing attacks.

## 2. Motivational Analysis

In this section, we present a general framework for sentence-level watermarking and analyze vulnerabilities inherent to its prefix-based design, which can weaken watermark signals under structural perturbations. We then provide empirical evidence that these perturbations commonly arise during paraphrasing, which motivates us to develop a method more

resilient to such attacks.

### 2.1. Sentence-Level Watermarking

Sentence-level watermarking is a class of LLM watermarking techniques that embed statistically detectable signals into sentence-level semantics rather than token-level patterns, thereby improving robustness against paraphrasing attacks (Hou et al., 2024a;b; Ren et al., 2024; Dabiriaghdam & Wang, 2025). These methods are typically *prefix-based*, meaning that the watermark signals in each sentence are conditioned on its preceding context. For instance, SemStamp (Hou et al., 2024a) derives a hash from the preceding sentence to define the 'green' region for embedding the subsequently generated sentence. During detection, it counts the number of sentences whose embeddings fall within the corresponding 'green' regions conditioned on their preceding sentences to identify watermark signals. We provide a more detailed discussion of related work in Appendix C.

Formally, we abstract the prefix-based sentence-level watermarking framework from the perspective of detection. The watermarking algorithm comprises a secret key $sk$, a pseudorandom function $\text{PRF}_{sk}(\cdot, \cdot) \to \{0, 1\}$ which indicates whether two inputs follow a pseudorandom relation based on $sk$, and an information extractor $\mathcal{E}(\cdot)$, which typically maps a text to a set of discrete values serving as a watermarking seed. Given a preceding context $\mathbf{X}_{n-1} = \{x_1, x_2, \cdots, x_{n-1}\}$, we can detect a watermark signal in $x_n$ if and only if the following condition is satisfied:

$$\text{PRF}_{sk}(\mathcal{E}(\Gamma(\mathbf{X}_{n-1})), \mathcal{E}(x_{n-1})) = 1 \qquad (1)$$

where $\Gamma(\cdot)$ selects a subset of content from the context $\mathbf{X}_{n-1}$. In most existing work, only the immediately preceding sentence is selected as context, i.e., $\Gamma(\mathbf{X}_{n-1}) = \{x_{n-1}\}$ (Hou et al., 2024a;b; Dabiriaghdam & Wang, 2025). Nevertheless, such a design can potentially cause the loss of watermark signals of $x_n$ if $x_{n-1}$ is modified into some other $x'_{n-1}$ due to sentence merges or splits, since $x_n$ is then less likely to satisfy the intended pseudorandom relations with $x'_{n-1}$ than with $x_{n-1}$. We define this modification as a *structural perturbation*, which includes a combination of sentence merges or splits during paraphrasing, as well as adversarial sentence insertion or deletion. For the same reason, detecting a watermark from $x'_{n-1}$ given $x_{n-2}$ may also be unlikely, ultimately causing a compounded loss of the signals from multiple sentences. These adverse effects may be further amplified when the watermarking method conditions on a larger context.

The compounded loss of watermark signals caused by such a structural perturbation presents a fundamental limitation for prefix-based sentence-level watermarking, particularly under paraphrasing attacks. We will analyze and quantify this effect in the following sections.

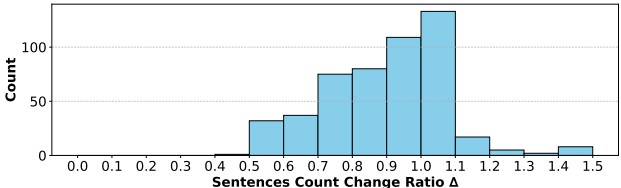

*Figure 1.* Sentence count change ratios $\Delta$ of GPT-3.5 paraphrases.

> **Before Paraphrasing**: Ruiz finished game two with three RBIs while Hanks and Diogen Ceballos each drove in two. Jose Mieses, Ceballos and ...

> **After Paraphrasing**: Ruiz recorded three RBIs in Game Two. Meanwhile, Hanks and Diogen Ceballos each contributed two RBIs. Jose Mieses ...

*Figure 2.* An example of paraphrases from GPT-3.5, which induces structural perturbation, where a sentence is split into two.

## 2.2. Empirical Analysis

Here, we conduct an empirical analysis to investigate whether the aforementioned structural perturbations commonly occur in paraphrasing. We use the C4 dataset (Kryscinski et al., 2022) and apply a widely used paraphrasing model in prior watermarking research: GPT-3.5 (OpenAI, 2022), to generate paraphrases of 500 random text samples. We then measure the change in text structure using a simple yet intuitive metric: the ratio of the number of sentences after paraphrasing to that before paraphrasing, denoted as $\Delta$. This metric effectively captures structural perturbations, as values of $\Delta$ unequal to 1 usually indicate splitting or merging of sentences during paraphrasing, which inevitably alters the context $x_{n-1}$ for some $x_n$.

The results are shown in Figure 1. It is evident that most paraphrases involve changes in sentence count, reflecting structural perturbations that commonly occur during paraphrasing. An illustrative example of sentence merging is provided in Figure 2.

In Section 4.2, we further observe that prefix-based methods degrade substantially under stronger paraphrasing. For instance, when using GPT-3.5 as the paraphraser, SemStamp's true positive rate at a 5% false positive rate (TPR@5%) drops from 91% to 22%, severely limiting its effectiveness. Similarly, in Section 4.3, its TPR@5% declines markedly to around 50% when we randomly delete only 20% of the existing sentences. These results motivate the development of a new method that is more robust to structural perturbations.

## 3. AliMark

Motivated by the analysis in Section 2, we propose *AliMark*, a sentence-level watermarking framework designed to alleviate the adverse impacts of text structural perturbations commonly appearing in paraphrasing. AliMark reformulates the sentence-level watermarking as a bit sequence encoding and alignment task to mitigate the adverse effects of structural perturbations, thereby improving the watermarking robustness under paraphrasing attacks.

Figure 3 presents an overview of AliMark. During embedding, AliMark selects each next sentence whose bit signals match its corresponding block of the secret bit sequence, such that the generated text encodes a sequence of bit signals. During detection, we generate multiple restructured text variants to account for potential sentence merges and splits, and adaptively align their extracted bit sequences with the secret bit sequence, aiming to identify the minimum alignment cost collectively. Detailed procedures are provided in Algorithm 1 and Algorithm 2.

### 3.1. Watermarked Text Generation

As discussed in Section 2, prefix-based sentence-level watermarking is vulnerable to signal degradation across multiple sentences under structural perturbations. Inspired by the observations in (Kuditipudi et al., 2024), we instead use a global secret bit sequence $\mathbf{s} \in \{0, 1\}^\infty$ as the watermark key, independent of any prefix context. The secret bit sequence $\mathbf{s}$ is partitioned into blocks of size $M$, with each block specifying the bit signals to be embedded in a single sentence. This design enables detection from a global perspective by aligning the bit signal patterns from the entire input text with the secret bit sequence, rather than relying on inter-sentence dependencies. Consequently, when structural perturbations arise during paraphrasing, more watermark signals can be captured, mitigating degradation and improving robustness.

Specifically, given the preceding context $\mathbf{X}_{n-1} = \{x_1, x_2, \ldots, x_{n-1}\}$, consisting of the prompt and optionally previously generated sentences, the LLM will first generate $Q$ different next sentences to form a candidate set $\mathbf{x}_n^*$. Our key objective is to select a sentence $x_n^q \in \mathbf{x}_n^*$ whose bit signals $\mathbf{b}^q(n) = \{b_{(n-1)M+1}^q, b_{(n-1)M+2}^q, \cdots, b_{nM}^q\}$ matches the corresponding $n$-th block of the secret bit sequence $\mathbf{s}(n) = \{s_{(n-1)M+1}, s_{(n-1)M+2}, \cdots, s_{nM}\}$, as the next sentence $x_n$.

To identify whether a candidate $x_n^q$ carries the desired bit signals, i.e., $\mathbf{s}(n)$, we compare its semantic embedding $\mathbf{e}_n^q$ against a set of pre-defined orthonormal secret vectors $\mathbf{V} = \{\mathbf{v}_1, \mathbf{v}_2, \ldots, \mathbf{v}_M\}$, each having the same dimensionality as the sentence embedding. The bit identification criterion $\varphi(\cdot, \cdot)$ is defined based on the sign of the inner product between the sentence embedding and a certain secret vector. Specifically, the $m$-th bit signal of the sentence $x_n^q$ is identified as:

$$b_{(n-1)M+m}^q \leftarrow \varphi(\mathbf{e}_n^q, \mathbf{v}_m) = \begin{cases} 0, & \text{if } \langle \mathbf{e}_n^q, \mathbf{v}_m \rangle < 0 \\ 1, & \text{otherwise} \end{cases} \quad (2)$$

For simplicity, we use $\Phi_{\mathbf{V}}(\cdot)$ to represent the bit signals extraction process, combining both the sentence embedding

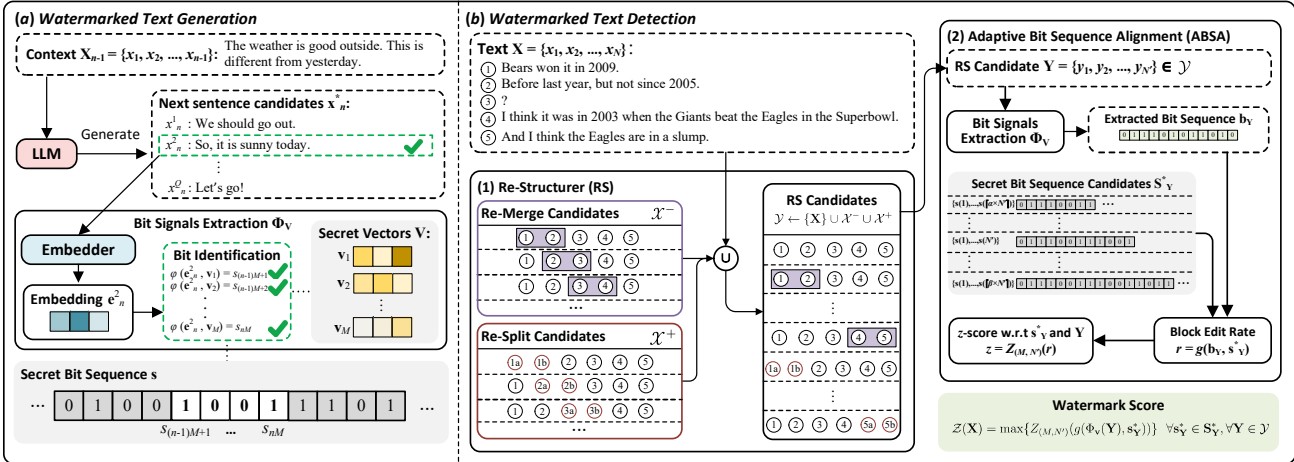

*Figure 3.* Overview of AliMark. **(a) Watermarked Text Generation**: Given the context, the LLM generates multiple candidates for the next sentence. The candidate whose extracted bit signals match the current block of the secret bit sequence is selected. **(b) Watermarked Text Detection**: The input text is proactively restructured into multiple candidate variants. Each variant is then converted into its corresponding bit sequence, and the alignment cost is computed against secret bit sequences of varying lengths. The maximum $z$-score across these attempts is taken as the final watermark score.

step and the bit identification step with secret vectors $\mathbf{V}$. Once we obtain the bit signals for each $x_n^q$ through $\Phi_{\mathbf{V}}(\cdot)$, the output next sentence $x_n$ is then selected randomly from candidates whose bit signals fully match the desired signals $\mathbf{s}(n)$ (i.e., with a matching count of $M$). If no such candidates exist, selection is made from those with the highest matching counts.

While we share similar approaches with SemStamp (Hou et al., 2024a) to transform sentence embedding into discrete values, a key distinction is that we do not use these values to determine a 'green' region for the next sentence. Instead, they directly function as the bit signals embedded within the current sentence, allowing the entire text to encode a sequential watermark and thereby enabling global-level sequence alignment for subsequent watermark detection.

### 3.2. Watermarked Text Detection

Given a text $\mathbf{X} = \{x_1, x_2, \cdots, x_N\}$ comprising $N$ sentences, the goal of watermark detection is to evaluate the likelihood that $\mathbf{X}$ encodes a bit sequence pattern derived from the secret bit sequence $\mathbf{s}$. We propose two complementary modules to address potential sentence splitting and merging during paraphrasing, as well as adversarial sentence insertion and deletion: (1) a *Re-Structurer (RS)* operating at the text level, which attempts to recover the original watermark structure by re-splitting and re-merging sentences; and (2) an *Adaptive Bit Sequence Alignment (ABSA)* module operating at the bit level, which aligns the bit signals of the re-structured texts to variable-length segments of $\mathbf{s}$. Both modules work collectively to mitigate the impacts of structural perturbations, which facilitates a robust search for the optimal alignment and strengthens the potential watermark

signals in paraphrases.

#### 3.2.1. RE-STRUCTURER

Since paraphrasing may occasionally split or merge sentences, the Re-Structurer (RS) module is designed to perform multiple re-merging and re-splitting attempts on the input text $\mathbf{X}$. If $\mathbf{X}$ originates from a watermarked text, some restructuring attempts may be able to recover the original watermarked text structure partially. This makes the bit signals of the restructured sentences identical or similar to their original counterparts, hence substantially increasing the watermark score of the entire text. In contrast, for unwatermarked texts (e.g., human-written text), these restructurings typically yield sentences with random bit signals, thus having a negligible impact on the watermark score.

In this work, we consider two simple re-merging and re-splitting operations, where the re-merging combines two consecutive sentences into a single sentence, while the re-splitting divides a sentence at its midpoint into two separate sentences.

Note that exhaustively enumerating all recursive combinations of re-merging and re-splitting is computationally infeasible due to combinatorial explosion (see Appendix A.1 for complexity analysis). Therefore, to ensure the RS module remains computationally tractable, we adopt a simplified strategy: restricting the RS module to perform a single-step operation. Specifically, for each pair of consecutive sentences $\{x_i, x_{i+1}\} \in \mathbf{X}$, we perform one re-merge operation, yielding $N-1$ re-merge candidates $\mathcal{X}^-$. Likewise, for each sentence $x_i \in \mathbf{X}$, we apply one re-split operation, producing $N$ re-split candidates $\mathcal{X}^+$. No additional re-structuring

attempts (i.e., multi-step re-merge and re-split operations) are considered.

By re-structuring the input text $\mathbf{X}$ into multiple re-structured variants, we form the RS candidate set $\mathcal{Y} \leftarrow \{\mathbf{X}\} \cup \mathcal{X}^- \cup \mathcal{X}^+$. Each candidate will be processed in the Adaptive Bit Sequence Alignment module to search for the lowest alignment cost to the secret bit sequence, as described in the following subsection.

### 3.2.2. ADAPTIVE BIT SEQUENCE ALIGNMENT

While the RS module partially alleviates sentence merging and splitting distortions caused by paraphrasing, it remains insufficient against other forms of perturbation. For example, if an adversary deletes several sentences from a water-marked text, the subsequent re-merging and re-splitting operations are unlikely to reconstruct the original watermarked structure, necessitating additional mitigation strategies.

To this end, we introduce the Adaptive Bit Sequence Alignment (ABSA) module, which further alleviates the adverse effects of these perturbations. At a high-level, for each RS candidate $\mathbf{Y} \in \mathcal{Y}$, ABSA computes the alignment costs between its extracted bit sequences $\mathbf{b_Y} \leftarrow \Phi_{\mathbf{V}}(\mathbf{Y})$ and a set of secret bit sequences of varying lengths. We define Block Edit Rate (BER), an adaptation of standard Levenshtein Distance tailored to our sentence-level manipulation scenarios, to quantify the alignment cost. This measure is then used to compute a $z$-score for each alignment attempt.

The motivation behind the ABSA module parallels that of the RS module. If $\mathbf{Y}$ originates from a watermarked text but still does not fully retain its original structure (e.g., in terms of sentence count), the adaptive alignment is likely to achieve a lower cost when using a secret bit sequence with a block count differing from the sentence count of $\mathbf{Y}$, resulting in a higher $z$-score. Otherwise, its effect is comparable to aligning two random bit sequences.

**Secret Bit Sequence Candidates**. Due to potential adversarial insertions or deletions of sentences, the number of blocks in the ground-truth secret bit sequence corresponding to the original watermarked text may deviate from $N'$. To accommodate this uncertainty, we construct a set of secret bit sequence candidates $\mathbf{S_Y^*}$ with varying numbers of blocks centered around $N'$:

$$\mathbf{S_Y^*} \leftarrow \{\{\mathbf{s}(1), \cdots, \mathbf{s}(\lceil \alpha \times N' \rceil)\}, \cdots, \\ \{\mathbf{s}(1), \cdots, \mathbf{s}(N')\}, \cdots, \{\mathbf{s}(1), \cdots, \mathbf{s}(\lceil \beta \times N' \rceil)\}\} \quad (3)$$

where $\alpha < 1$ and $\beta > 1$ are hyperparameters that control the lower and upper bounds on the number of blocks, respectively.

**Block Edit Rate**. Once we have the bit sequence $\mathbf{b_Y}$ and a secret bit sequence candidate $\mathbf{s_Y^*} \in \mathbf{S_Y^*}$, the next step is to evaluate the cost of aligning these bit sequences.

An intuitive way to calculate the cost is to use the Standard Levenshtein distance (Levenshtein et al., 1966), which operates at the bit level, assuming bit-wise independence of errors. However, in our sentence-level watermarking scenarios, the embedding unit is a sentence, which corresponds to a block of $M$ bits. Structure perturbations like sentence merges and splits affect the entire block simultaneously. Therefore, treating these errors as independent bit-level edits fails to capture their structural granularity.

Hence, we introduce *Block Edit Rate* (BER), which extends the standard Levenshtein distance to block-level operations. BER is derived from Block Edit Distance (BED), which measures the block-level alignment cost between $\mathbf{b_Y}$ and the selected $\mathbf{s_Y^*}$, and is normalized by the length of the longer sequence between $\mathbf{b_Y}$ and $\mathbf{s_Y^*}$. Like Levenshtein distance, BED is computed via a dynamic programming table $D$ over $\mathbf{b_Y}$ and $\mathbf{s_Y^*}$. In our framework, both the **(1) deletion of block** $\mathbf{b_Y}(i)$ and **(2) insertion of block** $\mathbf{s_Y^*}(j)$ incur a fixed cost equal to the block size $M$, whereas the **(3) substitution** cost between these two blocks is defined as their Hamming distance $h(\cdot, \cdot)$. As a result, the BED between the first $i$ blocks of $\mathbf{b_Y}$ and the first $j$ blocks of $\mathbf{s_Y^*}$ is:

$$D[i,j] = \min \begin{cases} D[i-1, j] + M \\ D[i, j-1] + M \\ D[i-1, j-1] + h(\mathbf{b_Y}(i), \mathbf{s_Y^*}(j)) \end{cases} \quad (4)$$

This design captures block-level manipulations while tolerating minor bit-level deviations induced by paraphrasing. After computing the entire DP table, BED is normalized to obtain BER. Algorithm 3 offers a complete procedure of BER calculation.

**Calculation of $z$-score**. Similar to most LLM watermarking methods (Kirchenbauer et al., 2023; Hou et al., 2024a), we compute a $z$-score to indicate the likelihood that a given text is watermarked or not. Here, we define the $z$-score as

$$Z_{(M,N')}(r) = \frac{\bar{\mu}_{(M,N')} - r}{\bar{\sigma}_{(M,N')}} \quad (5)$$

where $r = g(\mathbf{b_Y}, \mathbf{s_Y^*})$ represents the BER, while $\bar{\mu}$ and $\bar{\sigma}$ refer to the BER mean and standard deviation between two random bit sequences, respectively. Note that it is challenging to compute $\bar{\mu}$ and $\bar{\sigma}$ in closed form. Therefore, we estimate the values using Monte Carlo sampling based on different block sizes $M$ and the number of blocks $N'$. These precomputed estimations, denoted as $\bar{\mu}_{(M,N')}$ and $\bar{\sigma}_{(M,N')}$, are directly used in the $z$-score calculation. More estimation details are provided in Appendix A.2.

In practice, both the RS and ABSA modules can be further optimized through engineering refinements for computational efficiency; see Appendix A.3 for details.

*Table 1.* Watermarking performance comparison between baseline methods and AliMark using OPT-1.3B as the backbone LLM, evaluated under no attack and multiple paraphrasing attacks. All metrics are reported as percentages, with the best and second-best results highlighted in **bold** and underlined, respectively.

| Dataset | Method | No Attack | | | Pegasus | | | Parrot | | | DIPPER | | | GPT-3.5 | | |
|---|---|---|---|---|---|---|---|---|---|---|---|---|---|---|---|---|
| | | AUROC | TPR@1% | TPR@5% | AUROC | TPR@1% | TPR@5% | AUROC | TPR@1% | TPR@5% | AUROC | TPR@1% | TPR@5% | AUROC | TPR@1% | TPR@5% |
| Booksum | KGW | 100.0 | 100.0 | 100.0 | 55.2 | 1.0 | 1.6 | 53.6 | 0.4 | 0.6 | 49.8 | 0.2 | 0.2 | 46.4 | 0.0 | 0.0 |
| | EXP | 99.2 | 49.1 | 98.8 | 87.7 | 4.8 | 41.4 | 86.6 | 7.9 | 44.3 | 78.4 | 2.4 | 21.1 | 62.2 | 0.2 | 2.4 |
| | SynthID | 100.0 | 100.0 | 100.0 | 90.5 | 5.8 | 26.4 | 93.5 | 9.8 | 37.2 | 70.9 | 0.0 | 1.8 | 62.6 | 0.0 | 1.0 |
| | SIR | 99.4 | 14.6 | 93.6 | 35.8 | 0.0 | 0.4 | 28.0 | 0.0 | 0.6 | 35.9 | 0.0 | 0.6 | 32.9 | 0.0 | 0.2 |
| | SemStamp | 98.3 | 91.4 | 91.4 | 94.0 | 65.8 | 70.8 | 96.1 | 67.9 | 73.6 | 81.6 | 20.3 | 26.7 | 78.0 | 17.0 | 22.0 |
| | k-SemStamp | 100.0 | 98.2 | 98.5 | 96.2 | 60.5 | 62.8 | 97.3 | 68.5 | 70.2 | 78.4 | 12.7 | 14.1 | 78.1 | 15.3 | 16.6 |
| | SimMark | 99.5 | 70.8 | 91.0 | 99.0 | 45.8 | 76.2 | 99.1 | 43.6 | 78.0 | 87.9 | 6.4 | 23.4 | 85.0 | 6.6 | 18.2 |
| | PMark | 99.5 | 97.0 | 97.0 | 97.2 | 82.6 | 86.0 | 99.3 | 92.0 | 93.0 | 86.5 | 26.6 | 30.4 | 88.6 | 29.2 | 33.0 |
| | AliMark | 100.0 | 100.0 | 100.0 | 99.8 | 93.8 | 95.6 | 99.8 | 95.2 | 96.6 | 94.2 | 55.8 | 61.6 | 96.1 | 60.2 | 66.6 |
| C4 | KGW | 100.0 | 100.0 | 100.0 | 50.0 | 1.0 | 1.0 | 54.1 | 0.0 | 0.2 | 53.5 | 0.6 | 0.6 | 46.5 | 0.2 | 0.4 |
| | EXP | 99.8 | 99.6 | 99.6 | 88.2 | 47.5 | 51.9 | 89.2 | 51.1 | 54.3 | 76.7 | 21.0 | 24.8 | 64.6 | 6.2 | 8.2 |
| | SynthID | 100.0 | 100.0 | 100.0 | 92.7 | 36.8 | 46.6 | 91.6 | 43.6 | 51.4 | 70.9 | 5.8 | 8.4 | 66.1 | 3.4 | 5.4 |
| | SIR | 99.9 | 95.4 | 98.2 | 45.6 | 0.2 | 1.2 | 37.6 | 0.0 | 0.2 | 42.2 | 0.0 | 0.2 | 44.9 | 0.0 | 0.4 |
| | SemStamp | 97.8 | 84.2 | 84.9 | 92.7 | 47.1 | 53.5 | 93.0 | 53.4 | 56.9 | 79.1 | 14.7 | 17.9 | 75.2 | 12.7 | 16.8 |
| | k-SemStamp | 99.3 | 23.3 | 91.1 | 93.5 | 2.4 | 33.9 | 94.8 | 3.0 | 39.9 | 81.5 | 0.2 | 9.0 | 77.6 | 0.3 | 6.1 |
| | SimMark | 98.6 | 13.8 | 75.4 | 97.4 | 6.6 | 58.2 | 97.7 | 4.0 | 60.6 | 89.1 | 1.2 | 20.2 | 80.8 | 0.6 | 8.6 |
| | PMark | 98.9 | 94.8 | 95.0 | 96.1 | 78.6 | 79.6 | 98.1 | 88.6 | 89.4 | 84.2 | 28.6 | 29.6 | 85.1 | 26.4 | 28.2 |
| | AliMark | 99.9 | 99.2 | 99.4 | 98.2 | 69.6 | 81.2 | 98.9 | 83.6 | 91.2 | 91.5 | 37.0 | 49.8 | 92.3 | 36.0 | 51.6 |

*Table 2.* Watermarking performance comparison between baseline methods and AliMark using Qwen3-1.7B as the backbone LLM, evaluated under no attack and multiple paraphrasing attacks. All metrics are reported as percentages, with the best and second-best results highlighted in **bold** and underlined, respectively.

| Dataset | Method | No Attack | | | Pegasus | | | Parrot | | | DIPPER | | | GPT-3.5 | | |
|---|---|---|---|---|---|---|---|---|---|---|---|---|---|---|---|---|
| | | AUROC | TPR@1% | TPR@5% | AUROC | TPR@1% | TPR@5% | AUROC | TPR@1% | TPR@5% | AUROC | TPR@1% | TPR@5% | AUROC | TPR@1% | TPR@5% |
| Booksum | KGW | 100.0 | 99.5 | 99.6 | 40.9 | 0.8 | 0.8 | 46.7 | 0.0 | 0.0 | 41.7 | 0.2 | 0.2 | 50.1 | 0.0 | 0.0 |
| | EXP | 99.5 | 45.0 | 94.6 | 52.2 | 0.3 | 1.2 | 48.9 | 0.4 | 1.2 | 48.9 | 0.6 | 2.0 | 50.2 | 0.0 | 0.0 |
| | SynthID | 99.8 | 44.8 | 96.3 | 81.2 | 0.0 | 6.0 | 86.8 | 0.0 | 9.0 | 66.6 | 0.0 | 0.2 | 57.6 | 0.0 | 0.6 |
| | SIR | 98.1 | 27.2 | 68.8 | 88.3 | 3.8 | 23.2 | 94.0 | 3.8 | 34.6 | 77.7 | 0.2 | 6.8 | 72.6 | 0.2 | 2.8 |
| | SemStamp | 99.6 | 89.9 | 95.6 | 95.1 | 45.9 | 64.3 | 95.8 | 49.8 | 67.6 | 81.3 | 11.8 | 22.3 | 75.7 | 6.3 | 15.7 |
| | k-SemStamp | 99.6 | 93.8 | 94.7 | 93.1 | 46.3 | 48.8 | 94.9 | 54.0 | 57.3 | 76.2 | 13.6 | 14.7 | 74.5 | 9.8 | 12.2 |
| | SimMark | 96.8 | 35.8 | 65.4 | 95.0 | 16.2 | 41.6 | 95.5 | 15.6 | 44.2 | 86.5 | 2.0 | 14.4 | 75.5 | 0.6 | 4.4 |
| | PMark | 99.6 | 97.6 | 97.8 | 98.4 | 78.4 | 82.2 | 98.2 | 81.0 | 85.0 | 89.5 | 28.0 | 34.0 | 85.5 | 19.6 | 24.0 |
| | AliMark | 100.0 | 99.9 | 99.9 | 99.2 | 88.6 | 91.4 | 99.5 | 90.2 | 92.2 | 94.2 | 55.2 | 61.4 | 93.1 | 48.2 | 53.6 |
| C4 | KGW | 100.0 | 98.4 | 99.2 | 42.1 | 0.6 | 1.0 | 45.6 | 0.0 | 0.8 | 40.7 | 0.6 | 1.2 | 46.6 | 0.0 | 1.0 |
| | EXP | 99.4 | 97.6 | 98.0 | 51.3 | 2.0 | 2.0 | 51.0 | 1.0 | 1.4 | 50.5 | 0.6 | 1.8 | 50.4 | 0.0 | 0.0 |
| | SynthID | 99.9 | 98.0 | 98.2 | 83.4 | 15.4 | 27.6 | 87.6 | 23.2 | 32.6 | 64.5 | 1.6 | 3.4 | 62.4 | 3.2 | 5.4 |
| | SIR | 99.6 | 75.6 | 92.4 | 92.7 | 8.6 | 38.8 | 97.0 | 24.0 | 64.4 | 79.3 | 0.6 | 7.6 | 82.3 | 2.2 | 13.6 |
| | SemStamp | 99.3 | 89.5 | 91.9 | 94.1 | 48.5 | 53.0 | 96.1 | 53.4 | 59.9 | 79.6 | 16.2 | 19.9 | 77.0 | 12.4 | 14.5 |
| | k-SemStamp | 99.6 | 56.5 | 75.6 | 93.6 | 12.2 | 17.4 | 95.2 | 17.4 | 25.7 | 78.7 | 1.1 | 2.7 | 78.3 | 1.9 | 4.6 |
| | SimMark | 96.4 | 3.2 | 62.2 | 94.5 | 1.8 | 48.0 | 94.8 | 0.8 | 46.2 | 86.5 | 0.2 | 14.8 | 72.8 | 0.0 | 4.8 |
| | PMark | 99.5 | 97.4 | 97.4 | 97.6 | 82.2 | 84.2 | 98.1 | 83.0 | 84.4 | 85.2 | 38.4 | 39.6 | 81.9 | 25.2 | 26.8 |
| | AliMark | 99.9 | 99.2 | 99.6 | 98.7 | 73.2 | 84.6 | 99.0 | 78.4 | 87.8 | 92.5 | 41.4 | 56.2 | 90.4 | 29.2 | 44.4 |

### 3.2.3. WATERMARK SCORE CALCULATION

Building upon the RS and ABSA modules, we define the watermark score for an input text $\mathbf{X}$ as the maximum $z$-score obtained across all attempts from both modules:

$$\mathcal{Z}(\mathbf{X}) = \max\{Z_{(M,N')}(g(\Phi_{\mathbf{v}}(\mathbf{Y}), \mathbf{s}_{\mathbf{Y}}^*))\} \\ \forall \mathbf{s}_{\mathbf{Y}}^* \in \mathbf{S}_{\mathbf{Y}}^*, \forall \mathbf{Y} \in \mathcal{Y} \tag{6}$$

## 4. Experiments

We conduct extensive experiments to answer the following research questions: **[RQ1]**: Can AliMark improve the watermark detection performance over existing baselines under different paraphrasing attacks? **[RQ2]**: How does AliMark perform under deliberate text structural perturbations? **[RQ3]**: How does each component of AliMark affect its overall watermark detection performance? **[RQ4]**: How does AliMark affect the text quality when equipped with different LLMs?

### 4.1. Settings

**Datasets**. We evaluate watermark robustness primarily on two benchmark datasets: (1) Booksum (Kryscinski et al., 2022) and (2) C4 (Raffel et al., 2020). For each dataset, we randomly sample 500 text instances. The first sentence of each instance is used as the prompt to generate watermarked text, while the remaining content serves as a human text counterpart.

**Baselines**. We select four token-level watermarks, KGW (Kirchenbauer et al., 2023), EXP (Kuditipudi et al., 2024), SynthID (Dathathri et al., 2024), SIR (Liu et al., 2024), and four sentence-level watermarks, SemStamp (Hou et al., 2024a), k-SemStamp (Hou et al., 2024b), SimMark (Dabiriaghdam & Wang, 2025), PMark (Huo et al., 2026), as baselines for performance comparison.

**LLM Backbones**. Following (Hou et al., 2024a; Dabiriaghdam & Wang, 2025; Huo et al., 2026), our primary evalua-

tion of each watermarking algorithm is performed on OPT-1.3B (Zhang et al., 2022), while additional experiments are carried out on Qwen3-1.7B (Yang et al., 2025).

**Paraphrasers**. To evaluate the robustness of different watermarking methods, we employ four distinct paraphrasers: Pegasus (Zhang et al., 2020), Parrot (Damodaran, 2021), DIPPER (Krishna et al., 2023), and GPT-3.5 (OpenAI, 2022). We distinguish between the former two, which operate as sentence-to-sentence paraphrasers, and the latter two, which function as text-to-text models capable of generating more sophisticated rewrites with structural perturbations.

**Metrics**. We also follow (Hou et al., 2024a; Dabiriaghdam & Wang, 2025; Huo et al., 2026) to use three metrics, AUROC, TPR@1%, and TPR@5% to evaluate the effectiveness of each watermarking method.

More details about the experimental settings are provided in Appendix B.

### 4.2. RQ1: Effectiveness

To comprehensively evaluate the effectiveness of AliMark, we conduct the main experiments under both no-attack settings and attacks by all four paraphrasers. Table 1 and Table 2 present the watermarking performance comparisons between AliMark and baseline methods on the Booksum and C4 datasets, using OPT-1.3B and Qwen3-1.7B as backbone models, respectively. Overall, AliMark outperforms existing baselines across almost all settings. Notably, under strong paraphrasers DIPPER and GPT-3.5, AliMark maintains competitive TPR@5% scores. For example, using OPT-1.3B as the backbone on the Booksum dataset, it achieves TPR@5% scores of 61.6% and 66.6% under DIPPER and GPT-3.5, respectively. In contrast, the scores of other sentence-level baselines are reduced to below 30.4% and 33.0%. This demonstrates that AliMark more effectively mitigates the adverse effects of structural perturbations. Under weaker paraphrasers, although the performance gap narrows, AliMark maintains the highest detectability across most metrics. Similar trends are also observed when using Qwen3-1.7B as the backbone model, as shown in Table 2.

### 4.3. RQ2: Adversarial Robustness

To further assess the robustness of AliMark against adversarial attacks, we conduct probing experiments in which irrelevant sentences from the dataset are randomly inserted, or existing sentences are deleted, or reordered at rates ranging from 0.1 to 0.5 in the watermarked texts. This setting creates a controlled environment where the semantic content of each original sentence remains unchanged, and only structural perturbations are introduced. We compare AliMark with SemStamp and PMark, as these sentence-level baselines usually achieve strong performance in our main

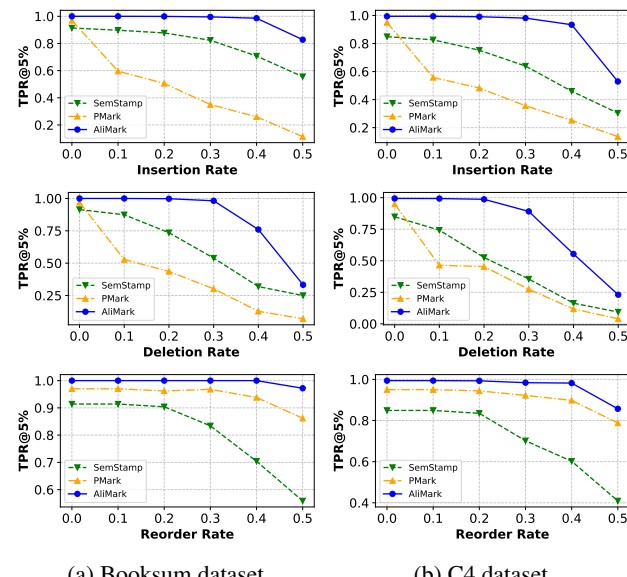

(a) Booksum dataset      (b) C4 dataset

*Figure 4.* Performance comparison of AliMark with baseline methods under three probing perturbation settings.

experiments. Evaluations are conducted on both the Booksum and C4 datasets, and performance is reported using TPR@5%.

The results are presented in Figure 4. Our primary observation is that AliMark exhibits substantially stronger robustness than competing methods under all probing attacks across most settings. For insertion and deletion, the largest performance gains are observed at perturbation rates between 0.2 and 0.4. Although performance degrades more noticeably at a rate of 0.5, such aggressive perturbations are likely to significantly distort the original semantics of the watermarked text, rendering them less realistic in practical scenarios. Under reordering perturbations, AliMark maintains strong detectability across a wide range of perturbation rates, whereas competing methods, particularly SemStamp, exhibit more pronounced performance degradation.

Overall, these results demonstrate that our design effectively mitigates the adverse effects of structural perturbations, leading to improved robustness.

### 4.4. RQ3: Module-wise Studies

**Impact of block size** $M$. We analyze the sensitivity of AliMark to the block size $M$, which determines the number of bits encoded per sentence. As shown in Figure 5, AliMark in general achieves superior performance with $M = 8$ compared to the other tested values across both datasets. The performance drop observed at $M = 2$ may be attributed to the lower information density per sentence, where only 4 possible values are insufficient to distinguish sentences, particularly under paraphrasing that may introduce devia-

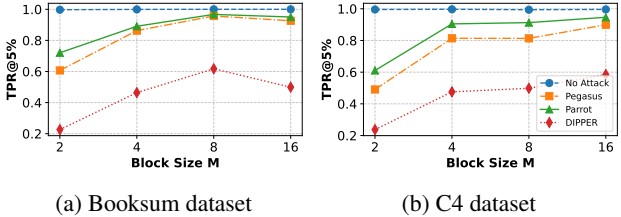

(a) Booksum dataset    (b) C4 dataset

*Figure 5.* Performance comparison of AliMark with different $M$.

*Table 3.* TPR@5% comparison of AliMark with different sentence embedders, evaluated under no attack and multiple paraphrasing attacks.

| Dataset | Sentence Embedder | No Attack | Pegasus | Parrot | DIPPER | GPT-3.5 |
|---|---|---|---|---|---|---|
| Booksum | multi-qa-mpnet-base-dot-v1 | 100.0 | 83.6 | 92.8 | 45.0 | 55.2 |
| | all-distilroberta-v1 | 99.8 | 93.6 | 93.2 | 53.6 | 56.8 |
| | all-mpnet-base-v2 | 100.0 | 95.6 | 96.6 | 61.6 | 66.6 |
| C4 | multi-qa-mpnet-base-dot-v1 | 98.2 | 80.8 | 89.4 | 46.2 | 50.6 |
| | all-distilroberta-v1 | 99.8 | 79.2 | 81.4 | 41.6 | 41.8 |
| | all-mpnet-base-v2 | 99.4 | 81.2 | 91.2 | 49.8 | 51.6 |

*Table 4.* TPR@5% comparison of AliMark with different next sentence candidate budget $Q$, evaluated under no attack and multiple paraphrasing attacks.

| Dataset | $Q$ | No Attack | Pegasus | Parrot | DIPPER | GPT-3.5 |
|---|---|---|---|---|---|---|
| Booksum | 8 | 97.4 | 53.2 | 62.4 | 25.4 | 29.6 |
| | 16 | 100.0 | 78.4 | 81.8 | 40.0 | 44.6 |
| | 32 | 100.0 | 88.8 | 92.2 | 52.7 | 61.2 |
| | 64 | 100.0 | 95.6 | 96.6 | 61.6 | 66.6 |
| C4 | 8 | 89.6 | 31.2 | 39.4 | 18.4 | 15.0 |
| | 16 | 98.0 | 58.4 | 69.8 | 26.8 | 28.2 |
| | 32 | 99.2 | 75.2 | 84.2 | 43.8 | 19.8 |
| | 64 | 99.4 | 81.2 | 91.2 | 49.8 | 51.6 |

tions in the bit signals. Conversely, increasing $M$ reduces the probability of random signal collisions exponentially $(1/2^M)$. The performance gains diminish as $M$ increases from 8 to 16. A plausible explanation is that, when $M$ becomes sufficiently large (e.g., $M = 16$), generating sentences that accurately align with the target signals becomes increasingly challenging.

**Impact of sentence embedder.** We also study the sensitivity of AliMark to the choice of sentence embedder. Specifically, we consider two alternatives, multi-qa-mpnet-base-dot-v1 (Song et al., 2020), and all-distilroberta-v1 (Sanh et al., 2020), which are selected to compare against our default choice, all-mpnet-base-v2 (Song et al., 2020). We evaluate AliMark with each embedder on both datasets. As reported in Table 3, the alternative embedders yield comparable performance, albeit slightly inferior to that of the default choice. Therefore, we retain all-mpnet-base-v2 as our default to ensure optimal semantic representations.

**Impact of next sentence candidate budget** $Q$. We further investigate the effect of the next sentence candidate budget $Q$ on the watermarking performance. Specifically, we consider four choices, including $Q = 8, 16, 32, 64$, with results presented in Table 4. It is clear that compared to our default

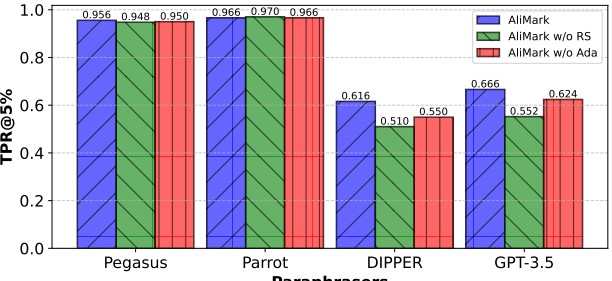

*Figure 6.* Performance comparison of AliMark variants.

*Table 5.* Runtime comparison across ablation variants with varying sentence counts.

| Method | Time (s) | | | | | |
|---|---|---|---|---|---|---|
| | $N = 4$ | $N = 8$ | $N = 16$ | $N = 32$ | $N = 64$ | $N = 128$ |
| AliMark | 0.02 | 0.03 | 0.05 | 0.08 | 0.15 | 0.34 |
| AliMark w/o RS | 0.02 | 0.02 | 0.02 | 0.03 | 0.05 | 0.07 |
| AliMark w/o Ada | 0.02 | 0.03 | 0.04 | 0.07 | 0.14 | 0.27 |

setting $Q = 64$, smaller budgets (e.g., 8, 16, 32) consistently yield inferior performance. Intuitively, a larger candidate budget increases the likelihood of achieving precise signal matching, thereby improving watermark detectability under both no attack and paraphrasing attack scenarios. While increasing $Q$ inevitably increases the generation overhead, we mitigate this computational overhead using the vLLM inference framework, since its optimized KV-cache design makes generating multiple next sentence candidates from the same prompt highly efficient (Kwon et al., 2023).

**Impact of detection modules.** Beyond analyzing the watermark embedding components, we conduct an ablation study to quantify the contribution of each component in the detection module. Specifically, we evaluate AliMark alongside two variants: (1) **w/o RS**, which removes the RS module, and (2) **w/o Ada**, which disables adaptive alignment over variable-length secret bit sequences (i.e., $\alpha = \beta = 1$). To assess effectiveness, we compare TPR@5% across these variants under different paraphrasers on the Booksum dataset. To assess efficiency, we measure the average runtime of these variants over 500 random texts with varying numbers of input sentences (ranging from 4 to 128).

Figure 6 shows the results on the Booksum dataset. For stronger paraphrasers like DIPPER and GPT-3.5, which are capable of causing significant structural perturbations, the contribution of each component becomes more pronounced. Notably, the removal of RS leads to a more substantial decline in detection rates compared to disabling adaptive alignment in most cases, highlighting the critical role of the restructuring mechanism in handling complex structural changes. In contrast, the performance gap between different variants is negligible for Pegasus and Parrot. This can be

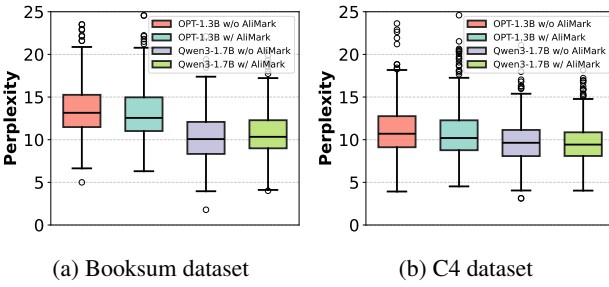

(a) Booksum dataset      (b) C4 dataset

*Figure 7.* Perplexity comparison between texts generated by different LLMs with and without AliMark.

*Table 6.* Perplexity comparison between texts generated by different LLMs with AliMark using different block size $M$.

| Dataset | Backbone | Block Size | | | |
|---|---|---|---|---|---|
| | | $M = 2$ | $M = 4$ | $M = 8$ | $M = 16$ |
| Booksum | OPT-1.3B | 12.9±3.2 | 13.0±3.2 | 13.2±3.3 | 12.8±3.1 |
| | Qwen3-1.7B | 10.6±2.8 | 10.5±3.0 | 10.7±2.6 | 10.8±2.5 |
| C4 | OPT-1.3B | 10.4±2.7 | 10.7±2.7 | 10.8±3.1 | 10.6±3.2 |
| | Qwen3-1.7B | 9.6±2.3 | 9.7±2.3 | 9.6±2.3 | 9.9±2.5 |

attributed to the fact that these models operate primarily as sentence-to-sentence paraphrasers, introducing limited structural perturbations. Consequently, the benefits of incorporating RS or ABSA are relatively limited.

AliMark also maintains low runtime cost during detection with different numbers of input sentences, as shown in Table 5. This is attributed to our engineering optimizations in the detection workflow (as described in Appendix A.3), which consolidate multiple DP table computations into a single computation per RS candidate. The detection time increases only modestly from AliMark w/o Ada to AliMark, rather than scaling multiplicatively. Fundamentally, the runtime overhead of detection is primarily driven by the number of RS candidates to be processed and their associated DP table computations. This dominance becomes more apparent as the number of input sentences increases. Although incorporating the RS module increases runtime, we consider this an acceptable trade-off given its greater improvements in watermark detection performance under strong paraphrasing attacks. We further analyze this trade-off and provide advice for future work in Section 5.

### 4.5. RQ4: Text Quality

To evaluate the impact of AliMark on text quality, we compute the perplexity (PPL) of texts generated by OPT-1.3B (Zhang et al., 2022), and Qwen3-1.7B (Yang et al., 2025), and compare them against unwatermarked LLM outputs. The PPL is calculated using a larger oracle Llama-3.1-8B (Grattafiori et al., 2024). As shown in Figure 7, our main observation is that AliMark produces perplexity distributions comparable to its unwatermarked counterpart, indicating that it introduces negligible degradation to text quality.

We further extend the text quality evaluation by analyzing multiple AliMark variants with different block sizes to examine their correlations. The results are reported in Table 6. We observe a slight increase in perplexity as $M$ increases. A plausible explanation is that a larger $M$ constrains sentence semantics to a narrower space, which may force the LLM to generate less natural text.

## 5. Limitations

While AliMark improves watermark robustness, particularly under strong paraphrasing, several limitations remain. First, the Re-Structurer (RS) module employs a single-step restructuring design and thus cannot handle more complex structural perturbations, such as multiple sentence merges and splits, thereby limiting the ability to achieve a higher watermark score. Second, RS generates $N - 1$ and $N$ candidates via re-merging and re-splitting, which can be inefficient in practice, as many variants are unnecessary. The resulting large number of bit sequence alignment attempts may also increase the runtime cost when $N$ is extremely large. Consequently, we anticipate that a more effective RS module, potentially utilizing a custom-trained model or more sophisticated heuristics, could adaptively determine which sentences should be merged or split. This will increase the likelihood of recovering the original watermarked structure before sequence alignment while simultaneously improving runtime efficiency – ultimately yielding more robust watermark detection under strong paraphrasing.

## 6. Conclusions

In this study, we demonstrate that prefix-based sentence-level watermarking methods are highly vulnerable to structural perturbations, such as sentence splitting and merging, which are commonly induced by advanced paraphrasers. To mitigate this issue, we propose AliMark, a reformulation of sentence-level watermarking as bit sequence encoding and alignment problems between a potentially watermarked text and a secret bit sequence. This design naturally improves robustness to sentence splitting and merging, as they usually result in a few offsets in the alignment. Extensive experiments show that AliMark significantly outperforms state-of-the-art baselines under strong paraphrasing attacks, including GPT-3.5 and DIPPER. Future work will focus on improving restructuring strategies to handle more complex sentence merging and splitting patterns efficiently.

## Acknowledgments

This research is supported by the Ministry of Education, Singapore, under the Academic Research Fund Tier 2 (FY2025) (Grant MOE-T2EP20124-0009), and NUS Artificial Intelli-

gence Institute (Grant A-8003888-00-00).

## Impact Statement

This paper proposes a sentence-level watermarking framework, AliMark, designed to improve robustness against structural perturbations induced by strong paraphrasing. The primary societal impact of this work lies in strengthening AI-generated text detection, thereby supporting copyright protection, content attribution, and academic integrity.

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

**Algorithm 1** Watermarked Text Generation with AliMark

**Input:** Context $\mathbf{X}_n = \{x_1, x_2, \cdots, x_{n-1}\}$, Secret bit sequence $\mathbf{s} = \{s_1, s_2, ...\}$, Block size $M$, Sentence embedder $\text{Emb}(\cdot)$, Secret vectors $\mathbf{V} = \{\mathbf{v}_1, \mathbf{v}_2, \ldots, \mathbf{v}_M\}$, Bit extractor $\varphi(\cdot, \cdot)$, Number of next sentence candidates $Q$

**Output:** Next sentence $x_n$

1: // Generate $Q$ next sentence candidates
2: $\mathbf{x}_n^* \leftarrow \text{LLM}(\mathbf{X}_{n-1}, \text{num\_return\_sentences} = Q)$
3: // Use $\Omega$ to record the bit matching counts for next sentence selection
4: Initialize array $\Omega = \{0, 0, \cdots, 0\}$ of size $Q$
5: // Select the next sentence matching the $n$-th block of secret bit sequence $s$
6: **for** $x_n^q \in \mathbf{x}_n^*$ **do**
7:    $\mathbf{e}_n^q \leftarrow \text{Emb}(\mathbf{x}_n^q)$ // Obtain sentence embedding
8:    **for** $m = 1$ **to** $M$ **do**
9:       $b_{(n-1)M+m}^q \leftarrow \varphi(\mathbf{e}_n^q, \mathbf{v}_m)$ // Extract the bit w.r.t secret vector $\mathbf{v}_m$
10:       **if** $b_{(n-1)M+m}^q = s_{(n-1)M+m}$ **then**
11:          $\Omega[q] \leftarrow \Omega[q] + 1$
12:       **end if**
13:    **end for**
14: **end for**
   // Select the next sentence randomly from the sentences with the highest bit matching counts
15: $x_n \leftarrow \text{SelectRandom}(\mathbf{x}_n^*, \Omega)$
16: **return** $x_n$

**Algorithm 2** Watermarked Text Detection with AliMark

**Input:** Text $\mathbf{X} = \{x_1, x_2, \cdots, x_N\}$, Secret bit sequence $\mathbf{s}$, Block size $M$, Secret vectors $\mathbf{V} = \{\mathbf{v}_1, \mathbf{v}_2, \ldots, \mathbf{v}_M\}$, Bit signals extractor $\Phi_{\mathbf{v}}(\cdot)$ for a text, Block Edit Rate calculator $g(\cdot, \cdot)$, $z$-score calculator $Z_{(M,N')}(\cdot)$

**Output:** A watermark score of Text $\mathbf{X}$

1: // (1) Re-Structurer
2: // Perform text-restructuring to $\mathbf{X}$ and obtain a set of RS candidates $\mathcal{Y}$
3: $\mathcal{X}^- \leftarrow \{\}, \mathcal{X}^+ \leftarrow \{\}$
4: **for** $i \in \{1, 2, \cdots, N-1\}$ **do**
5:    $x^- \leftarrow \text{MergeSentences}(x_i, x_{i+1})$
6:    $\mathcal{X}^- \leftarrow \mathcal{X}^- \cup \{\{\cdots, x_{i-1}, x^-, x_{i+2}, \cdots\}\}$
7: **end for**
8: **for** $i \in \{1, 2, \cdots, N\}$ **do**
9:    $x_a^+, x_b^+ \leftarrow \text{SplitSentence}(x_i)$
10:    $\mathcal{X}^+ \leftarrow \mathcal{X}^+ \cup \{\{\cdots, x_{i-1}, x_a^+, x_b^+, x_{i+1}, \cdots\}\}$
11: **end for**
12: $\mathcal{Y} \leftarrow \{\mathbf{X}\} \cup \mathcal{X}^- \cup \mathcal{X}^+$
13: $z_{\max} = -\inf$
14: // (2) Adaptive Bit Sequence Alignment
15: **for** $\mathbf{Y} \in \mathcal{Y}$ **do**
16:    // Extract bit sequence embedded in $\mathbf{Y}$
17:    $\mathbf{b_Y} \leftarrow \Phi_{\mathbf{v}}(\mathbf{Y})$
18:    // Calculate the number of blocks in $\mathbf{b}$
19:    $N' \leftarrow \frac{|\mathbf{b_Y}|}{M}$
20:    // Generate secret bit sequence candidates w.r.t. the $N'$
21:    $\mathbf{S_Y^*} \leftarrow \{\mathbf{s}_{[1:\lceil \alpha \times M \times N' \rceil]}, \cdots, \mathbf{s}_{[1:M \times N']}, \cdots, \mathbf{s}_{[1:\lceil \beta \times M \times N' \rceil]}\}$
22:    // Compute the $z$-score for $\mathbf{Y}$ w.r.t different $\mathbf{s}^*$
23:    **for** $\mathbf{s_Y^*} \in \mathbf{S_Y^*}$ **do**
24:       $r = g(\mathbf{b_Y}, \mathbf{s_Y^*})$
25:       $z \leftarrow Z_{(M,N')}(r) = \frac{\bar{\mu}_{(M,N')} - r}{\bar{\sigma}_{(M,N')}}$
26:       **if** $z > z_{\max}$ **then**
27:          $z_{\max} \leftarrow z$
28:       **end if**
29:    **end for**
30: **end for**
31: **return** $z_{\max}$

# A. Further Details of AliMark

The pseudocodes for watermarked text generation, watermarked text detection, and the calculation of block edit rate are provided in Algorithm 1, Algorithm 2, and Algorithm 3, respectively. A few examples of human texts, watermarked texts, and paraphrased texts are shown in Table 11.

## A.1. Complexity Analysis of the Re-Structurer

### A.1.1. Derivation of the Solution Space

We consider a text corpus consisting of $N$ initial sentences, denoted as $\mathbf{X} = \{x_1, x_2, \ldots, x_N\}$. The objective is to determine the number of unique re-structuring configurations achievable through at most $a$ merge operations and at most $b$ split operations, where $a, b < N$.

### A.1.2. The Separator Model

We model the text not as a sequence of sentences, but as a sequence of $2N$ atomic constituents. We assume each sentence $x_i$ can be split at exactly one valid internal point, dividing it into a left component ($L_i$) and a right component

($R_i$). The text sequence is thus represented as:

$$\mathcal{T}_{seq} = (L_1, R_1, L_2, R_2, \ldots, L_N, R_N) \qquad (7)$$

In this sequence, there are $2N - 1$ slots between adjacent components. These slots fall into two disjoint sets, corresponding to the two distinct operations:

**1. Boundary Slots (Merge Operations):** These are the positions between $R_i$ and $L_{i+1}$ for $1 \leq i < N$. There are $N - 1$ such slots.

- Initial State: A separator exists (representing the period between sentences).

- Action: Removing a separator corresponds to a *merge* operation.

**2. Internal Slots (Split Operations):** These are the positions between $L_i$ and $R_i$ for $1 \leq i \leq N$. There are $N$ such slots.

**Algorithm 3** Block Edit Rate Calculation $g$

**Input:** Bit sequence 1 $\mathbf{b}_1$, Bit sequence 2 $\mathbf{b}_2$, Block size $M$
**Output:** Block Edit Rate (BER) between $\mathbf{b}_1$ and $\mathbf{b}_2$

1: $N_1 \leftarrow \frac{|\mathbf{b}_1|}{M}$ // Number of blocks in $\mathbf{b}_1$
2: $N_2 \leftarrow \frac{|\mathbf{b}_2|}{M}$ // Number of blocks in $\mathbf{b}_2$
3: Initialize matrix $D$ of size $(N_1 + 1) \times (N_2 + 1)$
4: // Initialize boundary conditions
5: **for** $i = 1$ **to** $N_1$ **do**
6:     $D[i][0] \leftarrow i \times M$ // Cost of deleting $i$ blocks
7: **end for**
8: **for** $j = 1$ **to** $N_2$ **do**
9:     $D[0][j] \leftarrow j \times M$ // Cost of inserting $j$ blocks
10: **end for**
11: // Compute the dynamic programming table
12: **for** $i = 1$ **to** $N_1$ **do**
13:     **for** $j = 1$ **to** $N_2$ **do**
14:       // Delete the $i$-th block of $\mathbf{b}_1$
15:       $c_{\text{del}} \leftarrow D[i-1][j] + M$
16:       // Insert the $j$-th block of $\mathbf{b}_2$
17:       $c_{\text{ins}} \leftarrow D[i][j-1] + M$
18:       // Substitute bits between the $i$-th block in $\mathbf{b}_1$ (i.e., $\mathbf{b}_1(i)$) and the $j$-th block in $\mathbf{b}_2$ (i.e., $\mathbf{b}_2(j)$)
19:       $\mathbf{b}_1(i) = \{b_{1,(i-1)M+1}, b_{1,(i-1)M+2}, ..., b_{1,iM}\}$
20:       $\mathbf{b}_2(j) = \{b_{2,(j-1)M+1}, b_{2,(j-1)M+2}, ..., b_{2,jM}\}$
21:       $c_{\text{sub}} \leftarrow D[i-1][j-1] + \text{Hamming}(\mathbf{b}_1(i), \mathbf{b}_2(j))$
22:       // Calculate the block edit distance between the first $i$ blocks in $\mathbf{b}_1$ and the first $j$ blocks $\mathbf{b}_2$
23:       $D[i][j] \leftarrow \min(c_{\text{del}}, c_{\text{ins}}, c_{\text{sub}})$
24:     **end for**
25: **end for**
26: // Normalize the BED into BER
27: $r = \frac{D[N_1][N_2]}{\max\{|\mathbf{b}_1|, |\mathbf{b}_2|\}}$
28: **return** $r$

- Initial State: No separator exists (the sentence is continuous).

- Action: Inserting a separator corresponds to a *split* operation.

### A.1.3. COMBINATORIAL FORMULATION

Since the set of Boundary Slots and Internal Slots are disjoint, operations on them are mutually independent. The total number of variations is the product of the possibilities for merges and splits.

**Merge Possibilities** ($W_{\text{merge}}$). We are allowed to perform at most $a$ merge operations. This is equivalent to choosing $i$ boundary slots to remove, where $0 \leq i \leq a$. The number of ways to choose $i$ slots from $N - 1$ available boundaries is given by the binomial coefficient $\binom{N-1}{i}$. Summing over all valid $i$:

$$W_{\text{merge}} = \sum_{i=0}^{a} \binom{N-1}{i} \tag{8}$$

**Split Possibilities** ($W_{\text{split}}$). We are allowed to perform at most $b$ split operations. This is equivalent to choosing $j$ internal slots to activate (inserting a period), where $0 \leq j \leq$

$b$. The number of ways to choose $j$ slots from $N$ available internal positions is $\binom{N}{j}$. Summing over all valid $j$:

$$W_{\text{split}} = \sum_{j=0}^{b} \binom{N}{j} \tag{9}$$

**Total Configuration Count**. By the Multiplication Principle of combinatorics, the total number of distinct structural variations $\mathcal{T}(N, a, b)$ is:

$$\mathcal{T}(N, a, b) = W_{\text{merge}} \times W_{\text{split}} \tag{10}$$

Substituting the sums yields the final general formula:

$$\mathcal{T}(N, a, b) = \left( \sum_{i=0}^{a} \binom{N-1}{i} \right) \left( \sum_{j=0}^{b} \binom{N}{j} \right) \tag{11}$$

### A.1.4. ANALYSIS OF THE SOLUTION SPACE

Here, we consider the case where $a$ and $b$ are fixed constants independent of $N$ (i.e., $a, b \ll N$). Since $\binom{N}{k}$ represents a polynomial in $N$ of degree $k$, specifically $\binom{N}{k} \sim \frac{N^k}{k!}$, the partial sum is asymptotically dominated by its last term.

$$\sum_{i=0}^{a} \binom{N-1}{i} \sim \frac{N^a}{a!} = \Theta(N^a) \tag{12}$$

$$\sum_{j=0}^{b} \binom{N}{j} \sim \frac{N^b}{b!} = \Theta(N^b) \tag{13}$$

where $\Theta(\cdot)$ indicates a tight bound. Therefore, the total complexity grows polynomially:

$$\mathcal{T}(N, a, b) \approx \frac{N^{a+b}}{a!b!} = \Theta(N^{a+b}) \tag{14}$$

### A.1.5. PRACTICAL CONFIGURATION OF THE RE-STRUCTURER

Since the size of the search space grows polynomially in $N$, with degree determined by $a + b$, we only use two configurations $(a = 0, b = 1)$ and $(a = 1, b = 0)$ as our implementation choice for computational efficiency. This limits the complexity of the search space within $\Theta(N)$, which grows linearly with the sentence number $N$ only.

### A.2. BER Mean and Standard Deviation Estimation

The most accurate way to estimate the mean and standard deviation between two random bit sequences of different lengths is through Monte Carlo sampling using sequences of the corresponding lengths. However, enumerating all possible sequences of becomes computationally infeasible as the sequence length increases. To address this, we adopt

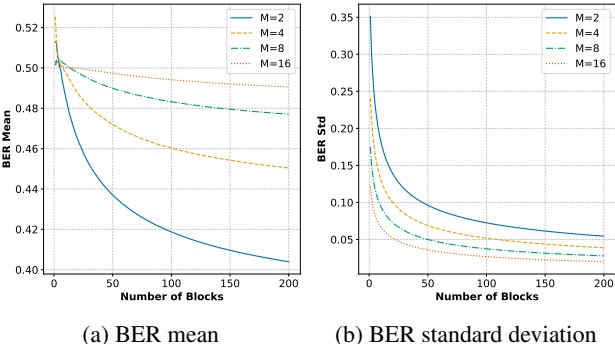

(a) BER mean      (b) BER standard deviation

*Figure 8.* Estimated mean and standard deviation of Block Edit Rate (BER) for varying block sizes $M$ and numbers of blocks $N'$.

a simplified approach: performing Monte Carlo sampling on equal-length random bit sequences to estimate these two quantities. This approximation generally underestimates the mean, since aligning with a longer or shorter sequence inevitably introduces block insertions and deletions, respectively. Conversely, it tends to overestimate the standard deviation, as the variation in alignment cost is limited to overlapping blocks. Collectively, such an estimation results in a slightly lower $z$-score, leading to a conservative prediction.

We estimate these two quantities using varying block sizes $M \in \{2, 4, 8, 16\}$ and numbers of blocks $N' \in \{1, 2, \cdots, 200\}$. For each combination of parameters, we randomly sample 1000 bit sequences of length $M \times N'$ and compute their empirical mean and standard deviation as our estimates. These values are visualized in Figure 8 and are directly utilized when calculating $z$-scores.

### A.3. Implementation Optimization

Our detection workflow, described in Figure 3 and Algorithm 2, may incur redundant computations in two procedures, which we discuss below and provide a more efficient implementation.

**(1) Sentence Embedding for RS Candidates**. The re-merging and re-splitting operations generate $N - 1$ and $N$ new sentences, respectively, resulting in an RS candidate set of size $1 + (N - 1) + N = 2N$. Following our workflow, this would require approximately $N$ sentence embedding operations per candidate, yielding a total complexity of $O(N^2)$.

This redundancy can be mitigated by embedding only the distinct sentences, rather than treating each RS candidate independently. Specifically, all RS candidates are constructed from the $N$ original sentences, $N - 1$ re-merged sentences, and $N$ re-split sentences. Therefore, embeddings need to be computed for only these $3N - 1$ sentences, reducing the overall complexity of sentence embedding to $O(N)$.

**(2) Bit Sequence Alignments**. Our workflow requires computing a dynamic programming (DP) table for each secret bit sequence candidate, which incurs a complexity of $O(N^2)$ per candidate. With $2N$ candidates, this would naively result in a total complexity of $O(N^3)$. However, since the longest secret bit sequence candidate encompasses all shorter candidates, it suffices to compute the DP table only for the longest candidate. The last row of this table can then be used to retrieve the block edit distances for the remaining candidates, which can be further used to obtain the BER. Consequently, the overall complexity is reduced to $O(N^2)$.

The runtime comparison reported in Table 5 was conducted using the two optimized implementations. The unoptimized workflow shown in Figure 3 and Algorithm 2 is retained solely for clarity of illustration.

## B. Further Details of Experiments

### B.1. Baselines

KGW (Kirchenbauer et al., 2023), EXP (Kuditipudi et al., 2024), SynthID (Dathathri et al., 2024), SIR (Liu et al., 2024), SemStamp (Hou et al., 2024a), and k-SemStamp (Hou et al., 2024b) are implemented using MarkLLM (Pan et al., 2024), an open-source LLM watermarking toolkit (`https://github.com/THU-BPM/MarkLLM`). Sim-Mark (Dabiriaghdam & Wang, 2025) and PMark (Huo et al., 2026) are implemented from `https://github.com/DabiriAghdam/SimMark` and `https://github.com/PMark-repo/PMark`, respectively. All methods use the default parameter configurations provided in their official codebases.

For AliMark, we use all-mpnet-base-v2 (Song et al., 2020) as the sentence embedder. We set the budget of next sentence candidates $Q$ to 64, the block size $M$ to 8. Each restructuring attempt in the RS module only splits one sentence or merges two consecutive sentences from the text. The secret bit sequence lower bound coefficient $\alpha$ and upper bound coefficient $\beta$ are set to 0.5 and 1.5, respectively.

### B.2. Text Generation

We adopt a fixed LLM generation configuration across all experiments, with top_p = 0.95, temperature = 0.7, and repetition_penalty = 1.15. For each prompt, we generate 12 sentences as LLM outputs.

### B.3. Watermark Detection

For watermark detection, we feed only the generated text into the detector, excluding the prompt from the input. This design choice reflects realistic detection scenarios, where the prompt is typically unavailable and, moreover, is not generated by the watermarking algorithm itself.

*Table 7.* Watermarking performance comparison between baseline methods and AliMark using gemma-3-1B-pt as the backbone LLM, evaluated under no attack and multiple paraphrasing attacks. All metrics are reported as percentages, with the best and second-best results highlighted in **bold** and underlined, respectively.

| Dataset | Method | No Attack | | | Pegasus | | | Parrot | | | DIPPER | | | GPT-3.5 | | |
|---|---|---|---|---|---|---|---|---|---|---|---|---|---|---|---|---|
| | | AUROC | TPR@1% | TPR@5% | AUROC | TPR@1% | TPR@5% | AUROC | TPR@1% | TPR@5% | AUROC | TPR@1% | TPR@5% | AUROC | TPR@1% | TPR@5% |
| Booksum | KGW | 100.0 | 100.0 | 100.0 | 60.9 | 0.2 | 0.2 | 56.0 | 0.0 | 0.0 | 63.3 | 0.2 | 1.0 | 58.3 | 0.0 | 1.8 |
| | SynthID | 100.0 | 100.0 | 100.0 | 91.7 | 5.6 | 44.8 | 96.3 | 9.8 | 60.8 | 76.4 | 0.0 | 6.6 | 65.2 | 0.0 | 2.6 |
| | SIR | 99.6 | 77.4 | 92.0 | 93.6 | 23.8 | 51.2 | 98.1 | 35.8 | 67.8 | 89.2 | 2.6 | 15.4 | 86.3 | 1.2 | 9.4 |
| | SemStamp | 99.7 | 98.4 | 98.4 | 97.6 | 72.7 | 78.2 | 98.3 | 77.5 | 83.1 | 86.4 | 31.2 | 36.5 | 77.1 | 12.8 | 18.0 |
| | k-SemStamp | 99.8 | 95.8 | 95.8 | 93.7 | 47.8 | 50.4 | 97.0 | 62.5 | 63.9 | 79.1 | 13.5 | 15.3 | 75.4 | 14.1 | 14.7 |
| | SimMark | 99.2 | 56.8 | 85.6 | 98.9 | 27.2 | 68.0 | 98.6 | 27.0 | 64.6 | 92.3 | 4.2 | 18.4 | 84.0 | 0.6 | 7.6 |
| | PMark | 99.1 | 94.0 | 94.2 | 97.1 | 76.6 | 80.2 | 98.3 | 86.8 | 88.8 | 86.3 | 23.2 | 27.8 | 88.3 | 24.4 | 31.0 |
| | AliMark | 100.0 | 100.0 | 100.0 | 99.7 | 89.5 | 93.3 | 99.8 | 96.1 | 97.6 | 94.1 | 52.1 | 56.1 | 95.4 | 59.1 | 65.3 |
| C4 | KGW | 100.0 | 100.0 | 100.0 | 76.4 | 1.6 | 2.2 | 72.0 | 1.0 | 2.0 | 79.9 | 5.0 | 7.4 | 77.1 | 2.8 | 5.2 |
| | SynthID | 100.0 | 100.0 | 100.0 | 91.7 | 31.6 | 43.6 | 95.9 | 45.0 | 56.4 | 73.4 | 1.4 | 5.4 | 68.4 | 3.2 | 7.0 |
| | SIR | 99.8 | 93.4 | 96.8 | 95.8 | 52.6 | 68.4 | 98.6 | 59.8 | 75.2 | 89.3 | 16.8 | 27.8 | 86.4 | 10.6 | 20.2 |
| | SemStamp | 99.6 | 96.0 | 96.8 | 93.7 | 48.4 | 53.8 | 96.1 | 53.9 | 59.6 | 78.9 | 13.7 | 18.0 | 75.7 | 10.7 | 15.3 |
| | k-SemStamp | 99.3 | 8.7 | 81.7 | 91.0 | 1.4 | 27.6 | 93.5 | 1.0 | 29.3 | 77.1 | 0.8 | 6.8 | 75.1 | 0.0 | 3.2 |
| | SimMark | 98.3 | 14.6 | 75.6 | 97.0 | 6.6 | 54.8 | 97.2 | 7.6 | 57.4 | 89.1 | 1.6 | 20.6 | 77.7 | 0.2 | 6.8 |
| | PMark | 98.4 | 93.0 | 93.2 | 94.7 | 69.6 | 71.4 | 97.1 | 81.2 | 82.0 | 83.3 | 22.8 | 24.0 | 81.4 | 20.4 | 23.0 |
| | AliMark | 99.9 | 99.2 | 99.8 | 98.4 | 65.7 | 81.0 | 99.2 | 75.7 | 88.2 | 91.6 | 33.3 | 49.9 | 91.9 | 34.0 | 50.9 |

*Table 8.* Watermarking performance comparison of baselines and AliMark with OPT-1.3B as the backbone LLM, evaluated under no attack and multiple paraphrasing attacks on the NQ dataset.

| Dataset | Method | No Attack | | | Pegasus | | | Parrot | | | DIPPER | | | GPT-3.5 | | |
|---|---|---|---|---|---|---|---|---|---|---|---|---|---|---|---|---|
| | | AUROC | TPR@1% | TPR@5% | AUROC | TPR@1% | TPR@5% | AUROC | TPR@1% | TPR@5% | AUROC | TPR@1% | TPR@5% | AUROC | TPR@1% | TPR@5% |
| NQ | KGW | 100.0 | 100.0 | 100.0 | 59.1 | 2.2 | 3.2 | 61.6 | 0.6 | 1.8 | 57.6 | 0.4 | 1.4 | 61.2 | 0.6 | 2.0 |
| | EXP | 98.0 | 96.4 | 97.0 | 55.6 | 0.8 | 2.6 | 57.2 | 1.2 | 3.0 | 59.2 | 2.4 | 4.6 | 55.6 | 0.2 | 0.8 |
| | SynthID | 100.0 | 100.0 | 100.0 | 86.4 | 6.2 | 14.0 | 86.9 | 8.2 | 18.2 | 57.9 | 0.0 | 0.0 | 55.0 | 0.0 | 0.0 |
| | SIR | 99.5 | 0.0 | 91.0 | 95.7 | 0.0 | 33.8 | 96.6 | 0.0 | 33.6 | 84.5 | 0.0 | 5.4 | 90.7 | 0.0 | 4.0 |
| | SemStamp | 99.7 | 94.1 | 97.0 | 97.6 | 58.0 | 79.4 | 97.0 | 41.4 | 70.8 | 84.5 | 10.3 | 24.7 | 79.5 | 5.2 | 17.6 |
| | k-SemStamp | 100.0 | 96.3 | 98.4 | 97.0 | 45.3 | 54.5 | 96.0 | 37.3 | 47.1 | 79.5 | 2.6 | 5.8 | 76.4 | 1.8 | 3.8 |
| | SimMark | 100.0 | 99.2 | 99.4 | 99.8 | 93.6 | 95.8 | 98.7 | 90.6 | 93.2 | 90.1 | 42.2 | 50.3 | 93.6 | 40.6 | 53.6 |
| | PMark | 98.6 | 92.2 | 94.6 | 95.8 | 48.0 | 79.0 | 97.1 | 47.0 | 81.8 | 77.1 | 5.6 | 17.6 | 75.9 | 5.4 | 16.2 |
| | AliMark | 100.0 | 100.0 | 100.0 | 99.9 | 93.8 | 97.2 | 99.9 | 93.7 | 96.9 | 91.7 | 49.5 | 55.2 | 95.1 | 54.7 | 59.8 |

---

**Prompt for GPT-3.5 Paraphraser**

You are a helpful assistant to rewrite text.
Please rewrite the following text, avoiding the use of the same words or phrases as the original text as much as possible. You are able to merge or split sentences, but must preserve the original meaning:
`{text}`

*Figure 9.* The prompt used in GPT-3.5 paraphraser

## B.4. Paraphrasers

For Pegasus and Parrot, we use the default configuration given by the SemStamp repository `https://github.com/abehou/SemStamp`, which performs paraphrasing on a sentence-to-sentence basis. For DIPPER (Krishna et al., 2023), we set lexical_diversity=60, order_diversity=0, sent_interval=1. For GPT-3.5 (OpenAI, 2022), we use the GPT-3.5-turbo model and adapt the prompt from (Huo et al., 2026) to enable sentence merging and splitting during paraphrasing (see Figure 9).

## B.5. Environment

All experiments were conducted on an Ubuntu server equipped with two Intel Xeon Platinum 8558 processors (48 cores each, 2.1 GHz) and four NVIDIA H200 GPUs with 140 GB memory each.

## B.6. Additional Results with Other LLM Backbones and Datasets

We conduct additional experiments to better understand how AliMark generalizes across different domains, including variations in LLM backbones and datasets. First, we adopt gemma-3-1B-pt (Team et al., 2025) as an alternative backbone and evaluate all watermarking methods on both the BookSum and C4 datasets. Second, noting that Book-Sum and C4 primarily involve sentence completion tasks, we further consider a question answering benchmark, Natural Questions (NQ) (Kwiatkowski et al., 2019), to assess the watermarking performance in a distinct domain. These results are shown in Table 7 and Table 8, respectively. Notably, AliMark continues to outperform baseline methods in these cross-domain settings, demonstrating its robustness and clear advantage over existing approaches.

## B.7. Impact of Complex Re-Structurer

We also study two alternative RS designs, including a multi-step re-structuring, which performs at most one re-merge and one re-split, and learned re-structuring, which performs re-structuring by prompting GPT-3.5 using the prompt from

**Prompt for Learned Re-structuring**

**Role**: You are an expert editor specializing in text clarity and logical structuring.
**Task**: I will provide you with a [Paraphrased Text]. During the paraphrasing process, the original sentence structures were distorted—multiple unrelated ideas were unnaturally merged into run-on sentences, and cohesive thoughts were inappropriately split.
Your goal is to restore this text to a natural, logical state.
**Instructions**:
1. Deconstruct and Split: Identify overly complex, merged sentences and split them into independent, atomic sentences, where each sentence conveys one clear core idea.
2. Merge and Cohere: Identify choppy, unnaturally split sentences and combine them back together to restore logical flow.
3. Preserve Meaning: Do not add any external information or remove core concepts.
4. Natural Transitions: Adjust conjunctions and punctuation to ensure the final output reads smoothly and professionally.
**[Paraphrased Text]**:
{paraphrased_text}
**Output format**: Please output the restored and logically structured text directly.

*Figure 10.* The prompt for learned re-structuring.

*Table 9.* TPR@5% comparison of AliMark with different re-structuring schemes, evaluated under no attack and multiple para-phrasing attacks.

| Dataset | RS scheme | No Attack | Pegasus | Parrot | DIPPER | GPT-3.5 |
|---|---|---|---|---|---|---|
| | multi-step | 100.0 | 95.0 | 96.4 | 59.4 | 66.4 |
| Booksum | learned | 77.4 | 69.8 | 67.2 | 46.8 | 56.0 |
| | ours | 100.0 | 95.6 | 96.6 | 61.6 | 66.6 |
| | multi-step | 97.6 | 76.2 | 87.4 | 43.2 | 43.4 |
| C4 | learned | 59.2 | 50.8 | 57.0 | 28.4 | 37.8 |
| | ours | 99.4 | 81.2 | 91.2 | 49.8 | 51.6 |

Figure 10. We evaluate this using OPT-1.3B as backbone, with the results reported in Table 9.

It is clear that neither approach outperforms our single-step design. For multi-step restructuring, a plausible reason for performance degradation may be that 2-step restructuring introduces significantly more trials of bit sequence alignments. For instance, a 12-sentence text yields 144 RS candidates under 2-step restructuring, compared to just 24 in our single-step approach. While this increased number of alignment trials can potentially increase the watermark score of a paraphrased text, it simultaneously inflates the scores of unwatermarked, human-written texts, making the two distributions less distinguishable.

Regarding learned restructuring, we observed a substantial performance drop even on unattacked texts, indicating that zero-shot restructuring with general-purpose LLMs is currently ineffective.

Given that these immediate alternatives prove to be both

*Table 10.* TPR@5% comparison of AliMark with different numbers of input sentences, evaluated under no attack and multiple paraphrasing attacks.

| Dataset | #Sentences | Method | No Attack | Pegasus | Parrot | DIPPER | GPT-3.5 |
|---|---|---|---|---|---|---|---|
| Booksum | 4 | SemStamp | 36.8 | 18.1 | 19.8 | 5.5 | 5.7 |
| | | PMark | 83.0 | 23.2 | 32.2 | 14.2 | 11.0 |
| | | AliMark | 96.8 | 44.4 | 48.4 | 28.6 | 29.2 |
| | 8 | SemStamp | 90.3 | 56.7 | 59.7 | 16.8 | 17.6 |
| | | PMark | 98.0 | 61.0 | 67.6 | 29.4 | 28.2 |
| | | AliMark | 100.0 | 65.6 | 74.6 | 33.4 | 39.6 |
| | 12 | SemStamp | 91.4 | 70.8 | 73.6 | 26.7 | 22.0 |
| | | PMark | 97.0 | 86.0 | 93.0 | 30.4 | 33.0 |
| | | AliMark | 100.0 | 95.6 | 96.6 | 61.6 | 66.6 |
| C4 | 4 | SemStamp | 52.3 | 25.9 | 30.3 | 9.8 | 7.9 |
| | | PMark | 88.6 | 24.4 | 34.0 | 15.0 | 15.2 |
| | | AliMark | 89.8 | 27.4 | 36.2 | 17.8 | 18.0 |
| | 8 | SemStamp | 85.0 | 50.3 | 54.8 | 17.9 | 16.1 |
| | | PMark | 95.8 | 68.6 | 76.0 | 35.2 | 23.8 |
| | | AliMark | 99.8 | 65.0 | 74.8 | 44.0 | 46.4 |
| | 12 | SemStamp | 91.9 | 53.0 | 58.9 | 19.9 | 14.5 |
| | | PMark | 95.0 | 79.6 | 89.4 | 29.6 | 28.2 |
| | | AliMark | 99.4 | 81.2 | 91.2 | 49.8 | 51.6 |

ineffective, we leave the development of a more advanced RS module, potentially utilizing a custom-trained model or more sophisticated heuristics, as a promising direction for future work.

### B.8. Impact of Input Sentence Count

Since the watermark detection fundamentally relies on statistical hypothesis testing over a sequence of elements, transitioning the carrier from tokens to sentences naturally reduces the available sample size. Consequently, diminished signal strength in short texts (e.g., 3–4 sentences) is an inherent challenge shared by all sentence-level watermarking paradigms.

To empirically assess this limitation, we conducted an additional evaluation using OPT-1.3B on both the Booksum dataset and the C4 dataset, comparing AliMark against SemStamp and PMark on shorter texts (specifically, sequences of 4 and 8 sentences). The results comparing TPR@5% are shown in Table 10. As anticipated, we observe a degradation in detectability and robustness across all methods as the sentence count decreases. Crucially, however, AliMark maintains a consistent and significant performance margin over the selected baselines. This demonstrates its superior watermarking performance, confirming its effectiveness even under heavily constrained input text lengths.

### B.9. Sentence Count Change Rate $\Delta$

We provide a more detailed analysis of the ability of different paraphrasers to induce structural perturbations by comparing watermarked texts generated using OPT-1.3B on the Booksum dataset with their paraphrases. As shown in Figure 11, sentence-to-sentence paraphrasers, including Pegasus and Parrot, are less likely to alter text structure, with only a small fraction of paraphrases exhibiting changes in sentence count. In contrast, DIPPER and GPT-3.5 produce

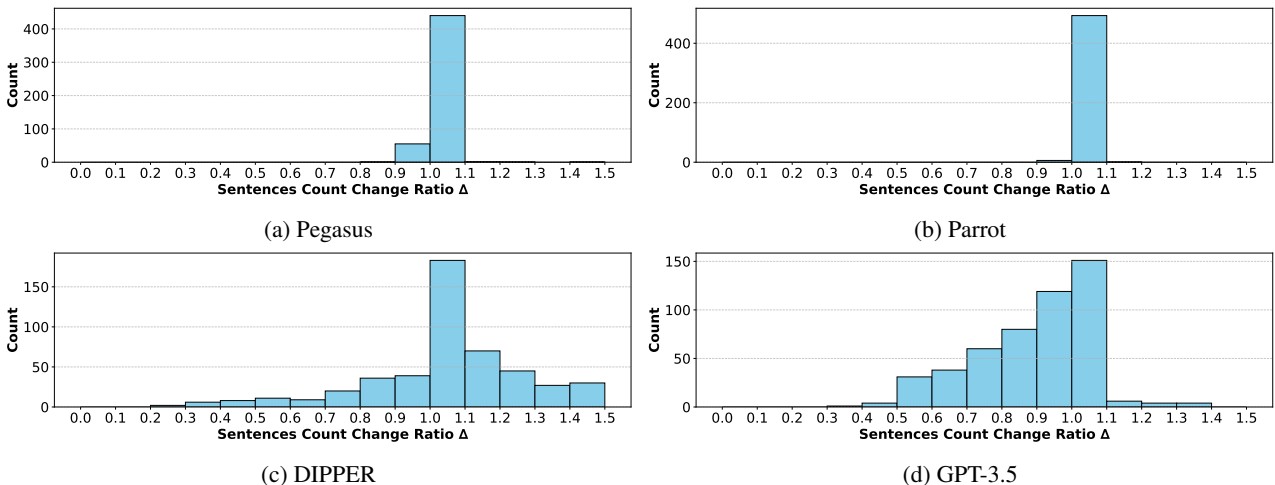

*Figure 11.* Sentence count change ratio (Δ) between the paraphrased and watermarked texts.

substantially stronger structural perturbations.

## C. Related Work

In recent years, LLM watermarking techniques have advanced rapidly. A seminal contribution in this area is KGW (Kirchenbauer et al., 2023), which embeds statistically detectable signals into text by introducing logit biases to pseudorandomly selected tokens, inspiring numerous subsequent studies. Here, we categorize LLM watermarking approaches into three main types: (1) zero-bit watermarking, (2) multi-bit watermarking, and (3) sentence-level watermarking.

**Zero-Bit Watermarking**. The primary objective of zero-bit watermarking is to determine whether a given text contains a watermark, with KGW being a representative method. Subsequent works optimize the KGW framework in watermark detectability and text quality. In terms of detectability, EWD enhances sensitivity in low-entropy contexts (e.g., code) by prioritizing high-entropy tokens through entropy-based z-score weighting (Lu et al., 2024). WinMax combats signal dilution in mixed texts using a sliding window approach to dynamically compute multi-window z-scores and select maxima, boosting robustness (Kirchenbauer et al., 2024). ITS and EXP further enhance detectability by embedding a secret key sequence into tokens and computing alignment costs during detection to mitigate adversarial token manipulations (Kuditipudi et al., 2024). Regarding text quality: existing methods focus on either embedding watermarks in high-entropy, low-semantic-impact positions using entropy or red-green list distributions (Wouters, 2024), optimizing list distributions via semantic alignment, dispersing similar words across lists (Chen et al., 2024), or increasing context-relevant word selection probability (Fu et al., 2024) to mitigate quality degradation. While the aforementioned methods focus on autoregressive LLMs, several recent stud-

ies have also investigated order-agnostic watermarking techniques (Chen et al., 2025a; Wu et al., 2025).

**Multi-Bit Watermarking**. Many applications require watermarks to convey richer information, such as user identifiers and timestamps, which zero-bit methods cannot support. Hence, another line of watermarking focuses on embedding a multi-bit message (i.e., a binary message containing several bits). Existing solutions to achieve this goal can be categorized into two main methods. The first splits the vocabulary into distinct groups (like color categories) to encode data bits (Fernandez et al., 2023). The second divides the text into chunks, with each chunk holding part of the data (Wang et al., 2024). However, if either method splits things too finely (e.g., too many groups or tiny chunks), the watermark becomes weak and harder to detect. To address this, (Yoo et al., 2024) merged both approaches, avoiding extreme splitting while maintaining enough data capacity. A most recent work empirically found that these three works suffer from either watermark decoding efficiency or detection accuracy when encoding more bits of the message, hence proposing a method to achieve a balance between these two characteristics by segmenting the watermark message (Qu et al., 2025a). Another work is designing robust mechanisms that can effectively trace content in scenarios involving many adaptive users who might attempt to evade detection. In parallel to these efficiency-focused optimizations, (Cohen et al., 2025) establishes rigorous statistical guarantees for ultra-low false positive rates, assessing the impact on model performance across standard benchmarks, and extending the mechanism to support multi-bit message embedding (Fernandez et al., 2023).

**Sentence-Level Watermarking**. However, the aforementioned methods operate mainly at the token level, making watermark signals vulnerable to adversarial rewriting.

Sentence-level semantic watermarking addresses this by embedding watermarks in semantic meaning, ensuring robustness against paraphrasing.

A representative work is SemStamp, which extends the idea of prefix-hashing from KGW into the sentence semantic space (Hou et al., 2024a). Specifically, it employs locality-sensitive hashing (Indyk & Motwani, 1998) to compute a hash from the preceding sentence, which defines the "green" region for embedding the subsequent sentence. During detection, SemStamp counts the number of sentences whose embeddings fall within the corresponding green regions, conditioned on their preceding sentences, and leverages this statistic to perform a test for watermark detection. A subsequent work, k-SemStamp (Hou et al., 2024b), mitigates the quality degradation caused by SemStamp's rigid restriction to a single semantic partition by allowing sampling from the $k$ nearest neighbor partitions, thereby improving fluency and semantic preservation. Similarly, SemaMark also relies on prefix hashing but adopts a Normalized Embedding Ring strategy (Ren et al., 2024). Beyond hash-based approaches, CoheMark (Zhang et al., 2025) and SimMark (Dabiriaghdam & Wang, 2025) explore more flexible watermarking criteria using fuzzy c-means clustering or sentence similarity. PMark (Huo et al., 2026) addresses a distorted sentence distribution problem observed in previous methods and proposes a median-estimation-based approach to achieve distortion-free sentence-level watermarking.

Our work falls within the domain of sentence-level watermarking but is primarily inspired by a token-level work (Kuditipudi et al., 2024), which mitigates the adverse effects of token insertion and deletion by embedding and aligning a global secret key sequence. The concept of correlating watermarks with a global secret key has been explored in a few prior works (Christ et al., 2023; Zhou et al., 2024; Fairoze et al., 2025; Huo et al., 2026); however, these approaches either operate at the token level or fail to maintain robustness under structural perturbations due to the exact bit-matching mechanism, ultimately remaining ineffective to defend against text-to-text paraphrasing attacks. Therefore, our work represents an enhancement to these existing approaches by reformulating the sentence-level watermarking as a bit sequence embedding and alignment problem with a multi-stage detection strategy to achieve better robustness.

Besides, both our method and existing multi-bit watermarking approaches embed multi-bit signals into text, but our objective is not to transmit a specific multi-bit message payload across the entire document. Instead, we assign varying multi-bit signals to individual sentences, such that the resulting bit sequence serves as evidence for a zero-bit watermark while simultaneously enabling post-hoc alignment.

## D. Broader Impact

A robust forensic audit approach, with robust LLM watermarking as a representative example, is fundamental to establishing accountability, protecting intellectual property, and ensuring security within rapidly evolving autonomous agent ecosystems (Xi et al., 2023; Han et al., 2026). In complex architectures where agents autonomously interact with Retrieval-Augmented Generation pipelines and automated code repositories, a resilient auditing framework allows creators and regulatory bodies to definitively trace whether proprietary information or copyrighted materials were unlawfully accessed, utilized, or leaked by the agent (Wang et al., 2026a;b). By integrating advanced provenance tracking approaches, techniques foundational to IP protection, organizations can maintain a verifiable and tamper-evident forensic trail of the agent's outputs, ultimately deterring intellectual property theft and ensuring traceability (Qu et al., 2025b; Guo et al., 2026). The necessity of such robust auditing is further underscored as generative capabilities expand into multimodal domains, where models face severe privacy risks from training data extraction and membership inference attacks (Zhai et al., 2024).

Furthermore, the same robust forensic logging paradigm that protects intellectual property significantly bolsters defenses against emerging agent-driven cyber threats. As autonomous agents are increasingly deployed to execute complex, real-world security tasks (Wang et al., 2025; Xu et al., 2026), they become prime targets for malicious exploitation. Alongside text-based threats such as indirect prompt injection (Greshake et al., 2023; Chen et al., 2025b; Cao et al., 2026b;a) or automated phishing campaigns (Li et al., 2024; Cao et al., 2025; Li et al., 2025), the integration of visual capabilities exposes these systems to multimodal data poisoning and backdoor attacks (Zhai et al., 2023). A robust auditing mechanism, thereby, serves as an indelible anchor that reliably links final outputs back to their generative origins or specific agent instances. This enables security analysts to rapidly identify compromised agent workflows, reconstruct attack vectors from manipulated environment inputs, and trace malicious directives back to their origin. Such a proactive oversight not only facilitates swift incident response and the immediate quarantine of rogue agents but also complements ongoing efforts to develop robust input-level defenses across modalities (Zhai et al., 2025), ensuring long-term trust and reliability in the deployment of autonomous AI technologies (Qu et al., 2026; de Witt et al., 2026).

*Table 11.* Examples of human text, watermarked text, and GPT-3.5 paraphrase of the watermarked text.

| | |
|---|---|
| **Prompt:** | Dr. W. H. R. Rivers, a psychiatrist at a mental hospital in Craiglockhart, Scotland, silently reads a letter written by Siegfried Sassoon in July 1917. |
| **Human Text**
score=-0.22 | Sassoon's declaration, a willful defiance of military authority, clearly and logically states his decision to stop fighting as a soldier in World War I. Sassoon believes that the purpose of the war has changed; what was once a war of liberation and defense has become one of aggression. He cites the suffering of the troops, the political insincerities, and the callous complacence of those at home as reasons for his protest. Rivers notices that Sassoon has signed the letter S. Sassoon, conveniently leaving out his first name, Siegfried. Bryce and Rivers discuss the possibility of Sassoon becoming a patient at the Craiglockhart hospital for shell-shocked soldiers. Rivers expresses his reservations, doubting that Sassoon is really shell-shocked and hesitant to shelter a coward who just wants to escape the fighting. Rivers is also concerned that bringing Sassoon to Craiglockhart will bring bad publicity to the hospital. Eventually, however, Bryce talks Rivers into taking Sassoon as a patient. Sassoon, on a train on his way to the mental hospital, thinks of a meeting he had with his friend Robert Graves a week earlier. In that meeting, Graves told Sassoon he had received his declaration protesting the war. Graves advises Sassoon to give up his cause and not to make a martyr of himself. Though Graves believes that the war has become unjust, he urges Sassoon to go to the mental hospital rather than allow himself to be court-martialed. Graves has pulled some strings; by showing the military board some of Sassoon's letters in which he writes of hallucinations of corpses in Piccadilly Circus in London. |
| **Watermarked Text**
score=3.91 | W. H. R. Rivers was born in 1885 in Glasgow and died there on March 28, 1951. (Click anywhere inside the image or below to swap between full screen and desktop versions). These letters are among many examples of correspondence sent during World War I from doctors working in Scottish hospitals. The collection is organized into three groups: medical and surgical, military and administrative. Images include letters from W. H. R. Rivers, William F. Drummond, A.J. Parker, E.C. Grant, Robert Murray, Bertram Maxwell, Lord John Sinclair, and Arthur Bowles Smith. This article was published in the SummerFall 2014 issue of the American Psychologist as an e-paper. AcknowledgmentsThanks go to J.R. Lewis for facilitating discussions about this project and to Eric Bayliss and Paul D. Shrum for their assistance in preparing the manuscript. There has been much controversy over the use of the word povertytoday2014the term has come to be associated with high unemployment rates, food shortages, and other problems. It used to mean just that; poverty existed and it caused suffering. But what do we mean when we say someone is poor? |
| **GPT-3.5 Paraphrases**
score=1.02 | Born in 1885 in Glasgow, W. H. R. Rivers passed away in the same city on March 28, 1951. Throughout World War I, many letters were exchanged among doctors in Scottish hospitals. The collection is divided into three categories: medical and surgical, military, and administrative. It contains correspondence from various individuals such as Rivers, Drummond, Parker, Grant, Murray, Maxwell, Sinclair, and Smith. An article in the SummerFall 2014 edition of the American Psychologist as an e-paper covered this topic. The project was supported by J.R. Lewis, Eric Bayliss, and Paul D. Shrum. There is ongoing debate surrounding the term povertyin modern times, associated with issues like unemployment and food shortages. While poverty historically indicated suffering, the current definition of being poor is subject to interpretation. |
| **Prompt:** | David receives such a large volume of mail because of his writing that he decides to have Traddles manage his correspondence from London. |
| **Human Text**
score=0.12 | In particular, David and Traddles discuss a letter that has arrived from Mr. Creakle, the former Salem House proprietor. He is now a magistrate who runs a model prison and the two young men decide to visit him. As Traddles and David are escorted through the building, Mr. Creakle explains that each prisoner is isolated so that they may all be restored to a wholesome state of mind, leading to sincere contrition and repentance.Mr. Creakle is very proud of two of his model prisoners, Numbers Twenty-Seven and Twenty-eight. David is amazed to find that they are Uriah Heep and Littimer! Uriah is in jail for fraud, forgery, and conspiracy, and when he sees David he sanctimoniously forgivesDavid for being violentto him and warns him to mend his ways. Littimer was imprisoned for robbing his master, and David learns that he would have escaped had it not been for Miss Mowcher, the dwarf hairdresser. David frequently visits Agnes to read her parts of his novel-in-progress. All the time he is with Agnes, he thinks of how much he loves her and what a perfect wife she would be. Shortly after Christmas, Aunt Betsey tells David that Agnes is about to be married. This rouses David to action, and he rides out to see Agnes to break down the barrier with a determined hand.Agnes is very reluctant to talk about her attachmentand she begins to cry. |
| **Watermarked Text**
score=4.14 | The letters are from people who love David's work and want to know more about him, so they write to him in the hope that he might be able to talk with them about their lives, their loves, or whatever else is on their minds. How many volumes do you think we will see? I would like to know when The Painted Desertgoes into production. Well, I don't really know since we haven't heard anything official yet. I am hoping for an early release! Thank you very much, dear readers, for your kind words and comments. Wednesday, November 20, 2013I'm back with another entry for the book tour! This time it comes from my friend Traddles in the UK. A short story by David HockingTraddles' blog:http:www.traddles-bryant.co.ukA little blurb:The third part of the epic trilogy set in a fictionalized version of Australia. Seventeen year old Violet Kelly has been left behind in Sydney after her parents move out. She finds herself working at the local pub and living alone until she meets Simon Sinek. He inspires her to start thinking differently, but she soon realizes that there isn2019t enough money to go around. |
| **GPT-3.5 Paraphrases**
score=1.46 | Letters have been received from admirers of David's art, expressing their desire to learn more about him. They reach out in the hopes of engaging in discussions with him about various topics such as their own experiences and interests. The number of volumes that will be published remains uncertain. As for the production timeline of The Painted Desert,I am eager to learn when it will begin. Unfortunately, there has been no official update on the matter yet. I am eagerly anticipating its early release! Your kind words and comments, dear readers, are greatly appreciated. On Wednesday, November 20, 2013, I return with another entry for the book tour. This one is shared by my friend Traddles from the UK. The blog can be found at http:www.traddles-bryant.co.uk. Here is a brief description: The narrative continues in a trilogy set in a fictional version of Australia. Violet Kelly, a seventeen-year-old, finds herself on her own in Sydney after her parents move away. Working at a local pub, she meets Simon Sinek, who challenges her to think differently. However, she soon realizes that there are financial constraints hindering her aspirations. |

