# OpenReview forum: "AliMark: Enhancing Robustness of Sentence-Level Watermarking Against Text Paraphrasing"
_ICML.cc/2026/Conference — ICML 2026 regular_

### Official Review · Reviewer_hRnv · 2026-03-07

**Soundness:** 2
**Presentation:** 3
**Significance:** 2
**Originality:** 3
**Overall Recommendation:** 3
**Confidence:** 4

**Summary:**

This paper studies the robustness limitations of existing sentence-level watermarking methods for text generation. While prior approaches improve resistance to paraphrasing by embedding watermarks at the sentence-semantic level, their prefix-based designs remain vulnerable to structural perturbations such as sentence splitting and merging, which commonly occur under strong paraphrasing systems (e.g., DIPPER or GPT-3.5). To address this limitation, the authors propose AliMark, a framework that reformulates sentence-level watermark detection as a bit-sequence encoding and alignment problem between the candidate text and a secret bit sequence. The method adopts a two-stage detection strategy: it first generates multiple restructured variants of the text and then adaptively aligns the extracted bit sequences with the secret sequence to minimize alignment cost.

**Compliance With Llm Reviewing Policy:**

Affirmed.

**Final Justification:**

Since my concerns are only partially addressed, I would keep my rating.

**Key Questions For Authors:**

See weaknesses.

**Limitations:**

Yes

**Strengths And Weaknesses:**

## Strengths

1. The proposed method looks novel. The authors provided detailed explanation and description about the watermarking algorithm, make it easy for the audience to understand.

2. The experiment is comprehensive, covering detectability, robustness, and generation quality.

## Weaknesses

1. The detector relys on multiple merge and split operations, which introduce heavy computational cost comparing to the one-pass detector (e.g. KGW).

2. The watermark signal exists at sentence level, which means that the watermark may not work well under low generation length, e.g., 100 token sequence with 3-4 sentences.

3. In table 1, the authors should also report the generation quality for each watermarking method. Normally, the watermark robustness grows with the watermark strength, and increasing watermark strength will decrease the generation quality. Without controlling the generation quality, it does not make sense to compare the watermark robustness.

---

> ### Author Rebuttal · Authors · 2026-03-31
>
> Dear reviewer hRnv,
>
> We deeply appreciate your constructive and insightful comment. We carefully respond to your concerns as follows.
>
> ## Q1: Text Quality Comparison with Baselines
>
> We agree that a fair watermarking robustness comparison should also consider text quality. As a result, we evaluate the text quality (measured by PPL) of watermarked texts generated by different watermarking methods using OPT-1.3B. The results are summarized in the table below. AliMark consistently maintains high text quality compared to most baselines. Hence, our watermarking method does not sacrifice text quality to achieve enhanced robustness.
>
> | Method/Dataset | Booksum   | C4            |
> |------------|---------------|---------------|
> | KGW        | 8.82±2.84     | 7.47±2.86     |
> | EXP        | 7.90±2.37     | 6.81±2.42     |
> | SynthID    | 9.51±2.79     | 8.71±2.40     |
> | SIR        | 9.85±2.20     | 9.45±2.28     |
> | SemStamp   | 8.86±2.47     | 7.34±2.82     |
> | k-SemStamp | 8.01±2.02     | 7.08±2.83     |
> | SimMark    | 8.99±2.77     | 8.67±3.91     |
> | PMark      | 5.77±1.99     | 5.01±1.89     |
> | AliMark    | 6.50±1.87     | 5.74±1.67     |
>
> ## Q2: Performance with Low Generation Length
>
> Since the watermark detection fundamentally relies on statistical hypothesis testing over a sequence of elements, transitioning the carrier from tokens to sentences naturally reduces the available sample size. Consequently, diminished signal strength in short texts (e.g., 3–4 sentences) is an inherent challenge shared by all sentence-level watermarking paradigms.
>
> To empirically assess this limitation, we conducted an additional evaluation using OPT-1.3B on the Booksum dataset, comparing AliMark against SemStamp and PMark on shorter texts (specifically, sequences of 4 and 8 sentences). The results comparing TPR@5% are shown below. As anticipated, we observe a degradation in detectability and robustness across all methods as the sentence count decreases. Crucially, however, AliMark maintains a consistent and significant performance margin over the selected baselines. This demonstrates its superior watermarking performance, confirming its effectiveness even under heavily constrained input text lengths.
>
> | #Sentences | Method   | No Attack | Pegasus | Parrot | DIPPER | GPT-3.5 |
> |------------|----------|-----------|---------|--------|--------|---------|
> | 4          | SemStmap | 39.6      | 19.0    | 22.4   | 8.4    | 5.6     |
> |            | PMark    | 76.8      | 20.8    | 31.8   | 12.6   | 8.8     |
> |            | AliMark  | 93.6      | 36.2    | 42.0   | 25.0   | 20.2    |
> | 8          | SemStamp | 95.6      | 60.2    | 64.4   | 24.8   | 16.8    |
> |            | PMark    | 95.2      | 51.8    | 65.0   | 28.0   | 17.6    |
> |            | AliMark  | 95.4      | 53.6    | 64.8   | 29.2   | 27.2    |
> | 12         | SemStamp | 96.6      | 73.8    | 79.2   | 34.4   | 20.0    |
> |            | PMark    | 95.2      | 74.6    | 76.2   | 29.6   | 17.4    |
> |            | AliMark  | 96.8      | 78.0    | 84.0   | 53.8   | 46.0    |
>
> ## Q3: Detection Runtime Cost
>
> Please refer to our response to Reviewer jVsu Q2 for more discussion about the efficiency of our detection module. Overall, we think that including RS is an acceptable trade-off to make, especially to defend against stronger paraphrasing attacks such as DIPPER and GPT-3.5.  We have also implemented several engineering optimizations to improve the efficiency of our detection module. Besides, if we encounter texts with an exceptionally large sentence count, we can also optionally bypass the RS module entirely, or by randomly sampling a specific ratio of the complete RS candidates, given that AliMark w/o RS also achieves strong performance compared to baselines.
>
> We hope our responses have addressed your concerns. If you have any additional questions or suggestions, we would be more than willing to discuss them during this review period. Finally, if you feel that our responses satisfactorily address your concerns, we would be sincerely grateful for your kind consideration in raising the score for our submission. Thanks again for your valuable reviews.
>
> Sincerely,
>
> The Authors of Submission 587

---

> > ### Author Rebuttal · Reviewer_hRnv · 2026-04-03
> >
> > 1. a) PPL is not "lower the better" measurement. b) SynthID and EXP are distortion-free watermarks and should be at similar PPL level, but it's not the case in the authors experiment.
> >
> > 3. Alimark is only slightly better than KGW, I don't think it achieved significantly better trade-off comparing to the KGW.
> >
> > I would keep my rating.

---

> > > ### Author Response · Authors · 2026-04-03
> > >
> > > Dear reviewer hRnv,
> > >
> > > Thanks for your feedback. We would like to address your new concern as follows:
> > >
> > > ## 1a. PPL is not "lower the better" measurement
> > >
> > > We agree with you. We selected PPL as our primary text quality metric mainly because most existing literature in this domain studied text quality using PPL as well. A potentially better way to evaluate the impact of a watermark using PPL is to measure the relative change in PPL compared to non-watermarked text. As illustrated in Figure 7 of our submission, the PPL distributions for texts generated by the non-watermarked and watermarked LLMs are comparable. For example, using OPT-1.3B on Booksum, the mean PPL of the watermarked text only increase 0.33 from their non-watermarked counterparts. Combined with the new empirical results provided in our earlier response, we conclude that AliMark maintains similar text quality on par with its non-watermarked counterpart while achieving significantly superior robustness against paraphrasing attacks compared to existing methods.
> > >
> > > We would like to also kindly ask for your advice on the alternative metric to evaluate text quality. We are more than willing to provide new results during the following discussion or in the final version.
> > >
> > > ## 1b. PPL Results of EXP and SynthID
> > >
> > > In our evaluations, the implementations of EXP and SynthID are directly adopted from a widely used LLM watermarking toolkit, MarkLLM (https://github.com/THU-BPM/MarkLLM). It is possible that the implementations of these two watermarking algorithms comprise some flaws, which affects its actual distortions. However, even if they are implemented perfectly, resulting in similar PPL, it still does not affect our main conclusions:
> > >
> > > - AliMark maintains similar text quality on par with its non-watermarked counterpart while achieving significantly superior robustness against paraphrasing attacks compared to existing methods.
> > >
> > > - AliMark does not deliberately sacrifice text quality to enhance watermarking robustness, meaning that our comparative analysis is fair.
> > >
> > > ## 2. Comparison of AliMark and KGW
> > >
> > > We would like to highlight several key observations from our experimental results that demonstrate AliMark's substantial advantages over KGW:
> > >
> > >
> > > - **Superior Robustness Across Paraphrasers**: As shown in Table 1, when using OPT-1.3B on the Booksum dataset, AliMark achieves a TPR@5% of **53.80%** and **46.00%** against DIPPER and GPT-3.5 paraphrasing attacks, respectively. In contrast, KGW achieves only 33.25% and 17.20%. This performance gap widens considerably on the C4 dataset, where AliMark yields **56.20%** and **47.00%**, compared to KGW’s 15.20% and 8.80%.
> > >
> > > - **Consistent Performance Across Models**: In Table 4 (page 15), we demonstrate that when using Qwen3-1.7B, KGW's TPR@5% drops consistently below 2% across all paraphrasing settings. Conversely, AliMark still maintains a high level of robustness. For instance, on the C4 dataset under DIPPER, AliMark retains a TPR@5% of **50.80%**, an over **36-fold improvement** compared to KGW.
> > >
> > > Collectively, these empirical observations confirm that AliMark provides a significant and robust leap in performance over KGW, rather than slight improvement.
> > >
> > >
> > > We hope that our new resnponse can address your concerns. Thanks again for your valuable comments.
> > >
> > > Sincerely,
> > >
> > > The Authors of Submission 587

---

### Official Review · Reviewer_jVsu · 2026-03-09

**Soundness:** 3
**Presentation:** 3
**Significance:** 3
**Originality:** 4
**Overall Recommendation:** 4
**Confidence:** 4

**Summary:**

The article keenly perceives the shortcomings of existing methods such as token level watermarking and sentence-level watermarking that are not resistant to "restructured text". A new sentence-level watermarking framework, AliMark, is designed for it, which utilizes a Re-Structurer to actively reconstruct sentences during detection. This paper defines the concept of Block Edit Rate for the first time, and a bit sequence alignment is adopted to calculate the block editing distance, thereby greatly improving the watermark's ability to resist rewriting attacks. Through comprehensive experiments, the author confirms that AliMark performs well on the standard dataset BOOKSUM, filling the gap in this field and having a positive impact on the current development of text watermark detection.

**Compliance With Llm Reviewing Policy:**

Affirmed.

**Final Justification:**

Thank you for the detailed rebuttal and for addressing my questions. After considering the overall discussion, I decided to maintain my original score.

**Key Questions For Authors:**

1.A necessary concern is whether the generation of watermarks by Alimark will lead to a great decrease in the quality of the generated text?
2.How to balance the cost loss caused by actively reconstruct sentence structures when detecting watermarks?
3.How to find the equilibrium point when selecting the range of coefficients values?

**Limitations:**

Yes

**Strengths And Weaknesses:**

Strengths:
1. The author possesses sharp insights. The paper points out that the current attack restructuring text poses a real threat to watermarks. Designing a Re-Structurer module to actively rewrite sentence structures is to solve such problems. AliMark's TPR@1 %FPR can reach 36-48%, which is a meaningful improvement.
2. The experimental design is clear and complete. Compare the experimental design with classic and more advanced watermarking methods. In different scenarios of no attack, weak attack, and strong attack, AliMark shows advantages, even particularly evident in DIPPER and GPT-3.5 strong attacks. Clearly demonstrated the effectiveness of AliMark.
3. The innovative concept of "Block Edit Rate" is proposed, which extends the Standard Levenshtein distance to the block level and matches with the structural granularity of word level watermarks. This will help promote subsequent research on sentence level watermarks and even AI generated text detection.

Weaknesses:
1. Firstly, there is a necessary concern about the cost of AliMark. As mentioned in the article, “RS generators N-1 and N candidates via re-merging and re-splitting... the runtime cost when N is extremely large.”A nice improvement cannot ignore time and economic costs, which may lose engineering significance.
2. It is extremely important to select the length of the secret sequence during detection, as it will affect the sentence deletion and insertion. “The secret bit sequence lower bound coefficient α and upper bound coefficient β are set to 0.5 and 1.5, respectively.” It provides a definite numerical value, and the situation of α=β=1 was also observed in the ablation experiment.
3. Another concern is the quality of the generated text. During reconstructing sentences to match watermarks, it is possible to cause a decrease in the quality of generated text. Please briefly show whether it will have an impact on fluent and semantic consistency.

---

> ### Author Rebuttal · Authors · 2026-03-31
>
> Dear reviewer jVsu,
>
> We deeply appreciate your constructive and insightful comment. We carefully respond to your concerns as follows.
>
> ## Q1: Impact of Restructuring to Generated Text Quality
>
> We believe there may be a misunderstanding. The restructuring process is not involved in the watermarked text generation phase; instead, it is only applied during the detection phase. Therefore, the restructuring process does not affect the quality of the generated watermarked text.
>
> Regarding the impact of AliMark on text quality, we provide a detailed evaluation in Figure 7 and Table 3, showing that watermarked texts exhibit comparable perplexity to those generated without watermarking. This indicates that AliMark does not introduce noticeable degradation in text quality.
>
> ## Q2: Cost of The Restructuring Module
>
> We agree with the reviewer that the Re-Structurer (RS) module introduces additional runtime overhead; however, we believe this is a highly acceptable trade-off for the resulting detection performance. As illustrated in Figure 6, incorporating the RS module yields significant performance gains when defending against strong paraphrasers such as DIPPER and GPT-3.5. Furthermore, the overall detection process remains efficient in practice. As demonstrated in Table 2, evaluating a 32-sentence text with the complete AliMark pipeline takes an average of only 0.66 seconds.
>
> To further address potential computational concerns, we would like to highlight the following structural and deployment advantages:
>
> - **Algorithmic Optimizations**: We have implemented several targeted engineering optimizations to improve the efficiency of both the RS and ABSA modules. As detailed in Appendix A.3, these refinements substantially reduce the computational complexity of both the sentence embedding and bit sequence alignment processes.
>
> - **Flexible Deployment**: For time-critical scenarios or texts with an exceptionally large sentence count, the detection time can be easily scaled down by optionally bypassing the RS module entirely, or by randomly sampling a specific ratio of the complete RS candidates. Importantly, as our evaluations demonstrate, even the strictly ablated version (AliMark w/o RS) consistently achieves significantly better performance than existing baseline methods, providing a flexible balance between speed and detectability.
>
>
> ## Q3: Equilibrium point of $\alpha$ and $\beta$
> We would like to first clarify that the length of secret bit sequence is not a fixed value, instead, it is determined by the sentence count of the input text. This machanism is scalable to texts with any sentence count. If the input text has $N$ sentences, we utilize secret bit sequences with lengths ranging from $\alpha N$ to $\beta N$.
>
> To better understand the optimal equilibrium for selecting the range coefficients, we conducted additional experiments using OPT-1.3B on the Booksum dataset, to evaluate how the choices of $\alpha$ and $\beta$ impact watermark detection performance. Specifically, we tested intermediate configurations of $\alpha=0.8, \beta=1.2$ and $\alpha=0.7, \beta=1.3$. The results comparing TPR@5% are shown in the table below, we observe that while expanding the range between $\alpha$ and $\beta$ initially improves detection performance, these gains saturate at the $\alpha=0.7, \beta=1.3$ configuration. A plausible explanation for this saturation is that achieving a lower bit sequence alignment cost with a significantly shorter or longer secret bit sequence is highly unlikely compared to sequences of comparable lengths. Consequently, an optimal equilibrium for these hyperparameters is to select values at the onset of this performance saturation area, avoiding the unnecessary computational overhead of setting an excessively small $\alpha$ or an excessively large $\beta$.
>
> | Configuration    | DIPPER | GPT-3.5 |
> |------------------|--------|---------|
> | $\alpha=1.0,\beta=1.0$ | 36.4   | 28.0    |
> | $\alpha=0.8,\beta=1.2$ | 51.2   | 44.6    |
> | $\alpha=0.7,\beta=1.3$ | 53.8   | 46.0    |
> | $\alpha=0.5,\beta=1.5$ | 53.8   | 46.0    |
>
> We hope our responses have addressed your concerns. If you have any additional questions or suggestions, we would be more than willing to discuss them during this review period. Finally, if you feel that our responses satisfactorily address your concerns, we would be sincerely grateful for your kind consideration in raising the score for our submission. Thanks again for your valuable reviews.
>
> Sincerely,
>
> The Authors of Submission 587

---

> > ### Author Rebuttal · Reviewer_jVsu · 2026-04-04
> >
> > Thank you for the detailed rebuttal. It addresses most of my earlier concerns.
> >
> > I still have one follow-up question. In the rebuttal, the extra results on α and β are helpful, but I am still not fully clear on how they connect to the default setting used in the paper. Since α and β directly control the adaptive alignment search range, this choice also seems related to the detection cost.
> >
> > I would still like to better understand why the paper keeps the current default range, and how the authors view the trade-off between robustness and efficiency here.

---

> > > ### Author Response · Authors · 2026-04-04
> > >
> > > Dear Reviewer jVsu,
> > >
> > > Thank you for your positive feedback and for continuing to engage with our rebuttal.
> > >
> > > To address your follow-up regarding our choice of default search range parameters ($\alpha=0.5, \beta=1.5$) over the 'equilibrium points' ($\alpha=0.7, \beta=1.3$) discussed earlier: our primary rationale for the wider default range is to guarantee more robust detection performance across a broader, more unpredictable spectrum of scenarios (e.g., under different paraphrasers/attacks). Besides, our default choice only yields a marginal detection overhead compared to $\alpha=0.7, \beta=1.3$ setting.
> > >
> > > To illustrate this, we first evaluated AliMark under two adversatial attack settings: randomly inserting unwatermarked sentences at a 50% ratio into the watermarked text (Insert@50%), and randomly deleting watermarked sentences at a 40% ratio from the watermarked text (Delete@40%). The results using OPT-1.3B on the Booksum dataset are detailed below:
> > >
> > > | Configuration    | Insert@50% | Delete@40% |
> > > |------------------|--------|---------|
> > > | $\alpha=1.0,\beta=1.0$ | 23.2   | 27.4    |
> > > | $\alpha=0.8,\beta=1.2$ | 35.8   | 36.2    |
> > > | $\alpha=0.7,\beta=1.3$ | 37.6   | 39.8    |
> > > | $\alpha=0.5,\beta=1.5$ | 40.2   | 42.2    |
> > >
> > > As it demonstrates, expanding the search range from [0.7, 1.3] to our default [0.5, 1.5] continues to yield performance gains under these adversarial attacks. This shows that the optimal configurations for $\alpha$ and $\beta$ may depend on the specific attack vector. We have to also admit that it is practically infeasible to tune them perfectly for every possible edge case.
> > >
> > > Regarding the trade-off between robustness and efficiency, we selected [0.5, 1.5] as the default because it provides a significant safety margin for robustness with acceptable computational overhead. For example, expanding the range from [0.7, 1.3] to [0.5, 1.5] increases the detection time by only 0.05 seconds (from 0.61s to 0.66s) for a 32-sentence text. We believe this minor efficiency cost is a highly favorable trade-off for the gains in adversarial resilience.
> > >
> > > We hope this clarifies our design choices. We would be grateful if you would consider raising your score should this resolve your final concern.
> > >
> > > Sincerely,
> > >
> > > The Authors of Submission 587

---

### Official Review · Reviewer_YMka · 2026-03-11

**Soundness:** 3
**Presentation:** 3
**Significance:** 2
**Originality:** 2
**Overall Recommendation:** 3
**Confidence:** 4

**Summary:**

This paper proposes AliMark, a sentence-level watermarking framework aimed at improving the robustness of LLM-generated text watermarks against paraphrasing attacks. The authors observe that existing sentence-level watermarking methods often rely on contextual dependencies between adjacent sentences, which can be disrupted by paraphrasing operations such as sentence splitting or merging, leading to detection failures.

To address this issue, the paper reformulates watermark detection as a bit sequence alignment problem, where each sentence encodes watermark bits derived from sentence embeddings and secret vectors. Detection aligns the extracted bit sequence with a reference sequence using dynamic programming, together with a restructuring module that generates candidate sentence structures. Experiments on Booksum and C4 under multiple paraphrasing attacks show improved robustness compared with several watermarking baselines.

**Compliance With Llm Reviewing Policy:**

Affirmed.

**Final Justification:**

The rebuttal provides useful clarifications and additional experiments, which improve the clarity of the paper.

However, my main concerns are only partially addressed. In particular, the methodological novelty remains limited, as the approach is still closely related to existing alignment-based watermark detection methods, without a clearly demonstrated new methodological contribution.

Therefore, my overall assessment remains unchanged, and I maintain my original recommendation.

**Key Questions For Authors:**

Please see weaknesses.

**Limitations:**

yes

**Strengths And Weaknesses:**

**Strengths**

- **Well-motivated problem.**
The paper identifies a practical limitation of existing sentence-level watermarking methods, namely their reliance on contextual dependencies between adjacent sentences, which can be disrupted by paraphrasing operations such as sentence splitting or merging.

- **Simple and intuitive formulation.**
Reformulating watermark detection as a bit sequence alignment problem provides a clear and intuitive way to handle structural perturbations in text.

- **Reasonable empirical evaluation.**
The method is compared with several token-level and sentence-level watermarking baselines under multiple paraphrasing attacks, demonstrating improved robustness.

**Weaknesses**

- **Limited novelty relative to prior watermark detection methods.**
While the bit sequence alignment formulation is reasonable, the core idea appears closely related to existing watermark detection approaches based on aligning or matching watermark signals. The contribution mainly lies in adapting this idea to the sentence-level setting.

- **Simplified restructuring strategy.**
The proposed Re-Structurer module only considers simple operations such as sentence splitting or merging. It is unclear how robust the method would be under more complex paraphrasing transformations (e.g., sentence reordering or multi-sentence rewriting).

- **Limited evaluation scope.**
Although several paraphrasing models are evaluated, the experiments do not include other realistic editing scenarios such as human paraphrasing or mixed editing operations.

---

> ### Author Rebuttal · Authors · 2026-03-31
>
> Dear reviewer YMka,
>
> We deeply appreciate your constructive and insightful comment. We carefully respond to your concerns as follows.
>
> ## Q1: Limited Novelty
>
> While sequence alignment has been explored in token-level methods, our core contribution lies in designing a novel paradigm specifically tailored to address a pervasive vulnerability in sentence-level watermarking: structural perturbations (e.g., sentence splitting and merging), which frequently appears in strong paraphrasing.
>
> Existing sentence-level methods rely on prefix-based ‘green/red’ region counting, which catastrophically fails when strong paraphrasers like GPT-3.5 alter sentence structures. To make sequence alignment viable and robust at the sentence level, our methodology introduces a new reformulation driven by two key innovations:
>
> - **Re-Structurer**: A proactive text-restructuring mechanism that mitigates structural perturbations. By attempting to recover the original watermark structure prior to alignment, it provides a highly effective, complementary pathway for detection that does not exist in token-level approaches.
>
> - **Block Edit Rate**: We introduce BER to specifically quantify the structural granularity and alignment costs of block-level burst errors. We further adopt BER with adaptive alignments against varying lengths of the secret bit sequence to search for lower alignment cost. This provides an effective way to search for lower alignment cost under structural perturbations.
>
> In short, our work is not a mere adaptation of existing alignment techniques, but a targeted methodological redesign necessary to make sentence-level watermarks robust against the structural perturbations commonly induced by advanced LLM paraphrasers.
>
> ## Q2: Sentence Reordering or Multi-Sentence Rewriting
>
> - **Sentence Reordering**. We evaluate the effects of sentence reordering on the Booksum dataset. Specifically, we randomly reorder the watermarked text with ratios ranging from 0.1 to 0.5. The results are provided below comparing the TPR@5%. We find that AliMark maintains stronger performance under such perturbations than other baselines, demonstrating robustness to sentence-level reordering.
>
> |Method/Reorder Rate|0.0|0.1|0.2|0.3|0.4|0.5|
> |-|-|-|-|-|-|-|
> |SemStamp|96.6|94.7|84.0|72.8|63.2|48.6|
> |PMark|95.2|95.0|94.2|91.6|89.0|73.2|
> |AliMark|96.8|96.8|96.8|96.4|95.2|93.8|
>
> - **Multi-sentence Rewriting**. Both GPT-3.5 and DIPPER rewrites at multi-sentence level instead of the single sentence, performing operations such as sentence merging and splitting across multiple sentences. This behavior is reflected in Figure 9 in our paper, where we report the sentence count change ratio between the watermarked and paraphrased texts. The ratio spans a wide range, indicating structural modifications to multiple sentences. For example, a paraphrase with 8 sentences and a sentence count change ratio of 0.8 corresponds to an original text with 10 sentences, implying that at least two sentences have been merged.
>
> Therefore, although our Re-Structurer module employs a single-step design, the use of GPT-3.5 and DIPPER in our evaluation already provides a strong testbed for robustness under complex, multi-sentence rewriting transformations. Notably, the results show that our single-step design remains effective and achieves strong performance under these challenging settings.
>
> ## Q3: Human Paraphrasing or Mixed Editing
>
> - **Alignment with Established Protocols**: To the best of our knowledge, existing research on LLM watermarking (e.g., [1][2]) did not employ human paraphrasing as an evaluation metric. Hence, our experimental setup is built on the established evaluation protocols in this domain to ensure a reproducible comparison with prior work.
>
> - **Practicality of the Threat Model**: In real-world scenarios, users aiming to remove watermarks typically rely on automated, machine-based paraphrasers rather than manual rewriting. The primary incentive for utilizing LLMs, such as a student generating an essay, is to maximize efficiency and save time. Manually rewriting the generated text introduces significant labor costs, effectively defeating the core purpose of using generative AI. Therefore, automated paraphrasing represents the most realistic, probable, and scalable attack vector.
>
> We hope our responses have addressed your concerns. If you have any additional questions or suggestions, we would be more than willing to discuss them during this review period. Finally, if you feel that our responses satisfactorily address your concerns, we would be sincerely grateful for your kind consideration in raising the score for our submission. Thanks again for your valuable reviews.
>
> Sincerely,
>
> The Authors of Submission 587
>
> **Reference**
>
> [1] PMark: Towards Robust and Distortion-free Semantic-level Watermarking with Channel Constraints. ICLR 2026.
>
> [2] SimMark: A Robust Sentence-Level Similarity-Based Watermarking Algorithm for Large Language Models. EMNLP 2025.

---

> > ### Author Rebuttal · Reviewer_YMka · 2026-04-04
> >
> > Thank you for the detailed rebuttal and additional clarifications.
> >
> > The authors provide helpful explanations regarding the design choices (e.g., Re-Structurer and Block Edit Rate) and include additional experiments such as sentence reordering. These clarifications improve the presentation and make the method easier to understand.
> >
> > However, my main concerns remain only partially addressed. In particular:
> >
> > (1) Novelty: While the reformulation is reasonable, the core idea still appears closely related to existing alignment-based watermark detection approaches. The rebuttal clarifies the motivation but does not fully demonstrate a clear methodological distinction beyond adapting alignment to the sentence level.
> >
> > (2) Evaluation scope: The additional experiments (e.g., sentence reordering) are helpful but remain limited. Evaluation under more realistic or diverse editing scenarios (e.g., human paraphrasing or mixed editing operations) is still missing, which leaves uncertainty about robustness in practical settings.
> >
> > Overall, while the rebuttal improves clarity, the remaining concerns relate to the core aspects of the work and cannot be fully resolved within the rebuttal period.

---

> > > ### Author Response · Authors · 2026-04-05
> > >
> > > Dear Reviewer YMka,
> > >
> > > Thanks for your feedback. We would like to provide a further discussion regarding both the methodological distinction and additional results on the reorder attack.
> > >
> > > ## 1. Methodological Distinction
> > > While we share a similar high-level philosophy of sequence alignment with token-level methods (e.g., EXP/ITS [1]), the physical reality of sentence-level paraphrasing necessitates fundamental methodological shifts. AliMark is not a trivial adaptation of Levenshtein distance to sentences; rather, it introduces a bespoke framework specifically designed to tackle structural perturbations that token-level methods cannot model. The core methodological distinctions lie in three key areas:
> > >
> > > **(a) Modeling Block-Level Errors via Block Edit Rate (BER)**:
> > >
> > > Token-level alignment methods assume independent, discrete bit-wise or token-wise errors, typically relying on the standard Levenshtein distance. However, in our sentence-level watermarking, the embedding unit is a sentence, which corresponds to a block of $M$ bits. Structural perturbations like sentence merging or splitting destroy the signal of the entire block simultaneously. To address this, treating these errors as independent bit-level edits fails to capture their structural granularity. Therefore, AliMark introduces the Block Edit Rate (BER). In our DP formulation, insertions and deletions are strictly executed at the block level (incurring a fixed cost of $M$), while the substitution cost between blocks is defined by their internal Hamming distance. This design specifically captures block-level manipulations while tolerating the microscopic bit-level deviations induced by paraphrasing.
> > >
> > > **(b) Dual-Level Alignment Architecture (Text-Level + Bit-Level)**:
> > >
> > > Existing alignment approaches operate exclusively in a single discrete space (e.g., aligning extracted sequences of hashes or tokens). This single-level approach fails at the sentence level because structural attacks (splits/merges) fundamentally alter the underlying semantic embeddings of the sentences, corrupting the extracted bit signals before alignment even begins. To resolve this, AliMark employs a Re-Structurer (RS) module that operates directly at the text level. It proactively generates multiple re-merged and re-split text variants to recover the original watermark structure. This 'text-level physical restructuring followed by bit-level sequence alignment' paradigm is a methodological innovation absent in purely token-level alignment frameworks.
> > >
> > > **(c) Adaptive Alignment Against Text-Length Distortion**:
> > >
> > > Token-level methods generally align a perturbed text against a fixed-length reference key. Under advanced paraphrasing attacks (e.g., using GPT-3.5 or DIPPER), adversaries frequently merge, split, insert, or delete sentences, leading to situations where the number of blocks in the ground-truth secret bit sequence corresponding to the original watermarked text may deviate from the current sentence count. AliMark’s Adaptive Bit Sequence Alignment (ABSA) module dynamically constructs a set of candidate secret bit sequences of varying lengths, centered around the current text length. This adaptive search allows AliMark to search for stronger watermark signals even when the global text length undergoes severe structural distortion. The necessity of these specific methodological advancements is empirically validated in our ablation study (Section 4.4). We observe that disabling the adaptive alignment (AliMark w/o Ada) leads to substantial performance declines under strong paraphrasing like DIPPER and GPT-3.5, proving that a fixed-length alignment mechanism alone is insufficient.
> > >
> > > To the best of our knowledge, [1] remains the sole work investigating sequence alignment for robustness in token-level watermarking. Although a limited number of other token-level studies, such as [2], utilize sequence alignment, their focus is on defending against spoofing attacks instead of achieving robustness.
> > >
> > > ## 2. Reorder Attack
> > > We provide new results using OPT-1.3B on the C4 dataset. As shown in the table below, AliMark still consistently outperforms baselines.
> > >
> > > |Method/Reorder Rate|0.0|0.1|0.2|0.3|0.4|0.5|
> > > |-|-|-|-|-|-|-|
> > > |SemStamp|98.0|96.8|73.6|59.0|44.0|33.2|
> > > |PMark|93.0|93.0|90.8|86.6|82.2|58.0|
> > > |AliMark|97.4|97.4|97.2|96.0|94.2|82.0|
> > >
> > >
> > > Sincerely,
> > >
> > > The Authors of Submission 587
> > >
> > > **Reference**
> > >
> > > [1] Robust Distortion-free Watermarks for Language Models. TMLR 2024.
> > >
> > > [2] Bileve: Securing Text Provenance in Large Language Models Against Spoofing with Bi-level Signature. NeurIPS 2024.

---

### Official Review · Reviewer_7gnv · 2026-03-11

**Soundness:** 3
**Presentation:** 3
**Significance:** 2
**Originality:** 3
**Overall Recommendation:** 4
**Confidence:** 3

**Summary:**

This paper studies a key weakness of sentence-level watermarks: although they are more robust than token-level methods against lexical paraphrasing, they still rely on prefix-conditioned sentence structure, and thus break badly when paraphrasers split or merge sentences. The paper proposes AliMark, which reformulates sentence-level watermarking as a bit-sequence encoding and alignment problem. It uses a global secret bit sequence for embedding, then applies a Re-Structurer and Adaptive Bit Sequence Alignment with a block-level edit rate during detection. Experiments show clear gains over prior methods, especially under stronger paraphrasers such as DIPPER and GPT-3.5.

**Compliance With Llm Reviewing Policy:**

Affirmed.

**Key Questions For Authors:**

How much would AliMark improve if the restructuring module allowed multi-step or learned restructuring, rather than only single-step edits?

How sensitive is the method to the choice of block size M, candidate count Q, and the sentence embedding model?

**Limitations:**

yes

**Strengths And Weaknesses:**

Pros

The paper identifies a real and under-addressed failure mode of sentence-level watermarking: structural perturbation, not just lexical paraphrasing. The motivation is strong and empirically supported.

The method is conceptually clean. Reformulating detection as global bit-sequence alignment is a meaningful departure from fragile prefix-based sentence matching.

The empirical gains are convincing under stronger attacks, and the improvement is especially meaningful where prior sentence-level methods degrade sharply.

Weaknesses

The Re-Structurer is restricted to single-step re-merge/re-split operations, so the method may still struggle when paraphrasing induces more complex multi-step structural changes.

Embedding requires generating Q candidate next sentences and selecting by bit matching, which likely adds noticeable generation overhead and may reduce practicality at scale.

The evaluation is still concentrated on Booksum/C4 and a small set of backbone models, so cross-domain generalization is not fully established.

---

> ### Author Rebuttal · Authors · 2026-03-31
>
> Dear reviewer 7gnv,
>
> We deeply appreciate your constructive and insightful comment. We carefully respond to your concerns as follows.
>
> (Remarks: All tables below report the TPR@5% metric.)
>
> ## Q1: Multi-step/Learned Restructuring
>
> Using OPT-1.3B on Booksum, we compared our single-step design against multi-step (one re-merge + one re-split) and learned restructuring (prompting GPT-3.5).
>
> | RS scheme | No Attack | Pegasus | Parrot | DIPPER | GPT-3.5 |
> |-|-|-|-|-|-|
> | multistep | 96.0 | 77.4 | 83.6 | 51.8 | 45.8 |
> | learned | 74.8 | 56.8 | 58.4 | 40.8 | 38.6 |
> | ours | 96.8 | 78.0 | 84.0 | 53.8 | 46.0 |
>
> Neither approach outperforms our single-step design. For multi-step restructuring, a plausible reason of performance degradation may be that 2-step restructuring introduces signanificant more trials of bit sequence alignments. For instance, a 12-sentence text yields 144 RS candidates under 2-step restructuring, compared to just 24 in our single-step approach. While this increased alignment trials can potentially increase the watermark score of a paraphrased text, it simultaneously inflates the scores of unwatermarked, human-written texts, making the two distributions less distinguishable.
>
> Regarding learned restructuring, we observed a substantial performance drop even on unattacked texts, indicating that zero-shot restructuring with general-purpose LLMs is currently ineffective.
>
> Given that these immediate alternatives prove to be both ineffective, we leave the development of a more advanced RS module, potentially utilizing a custom-trained model or more sophisticated heuristics, as a promising direction for future work.
>
> ## Q2: Sensitivity of Block Size $M$, Candidate Count $Q$, and Sentence Embedder
>
> We evaluated hyperparameter impacts using OPT-1.3B on Booksum:
>
> - **Block Size $M$**: The results have been already reported in Figure 5. We observed that increasing $M$ generally improves watermarking robustness. However, if $M$ is too large, it is becomes challenging to generate a sentence matching the target signals.
>
> - **Candidate Count $Q$**: Compared to our default $Q=64$, smaller pools (8, 16, 32) perform worse. A larger pool increases the chance of perfect signal matching. We mitigate computational overhead using vLLM inference framework, since its optimized KV-cache design makes generating multiple candidates from the same prompt highly efficient.
>
> | Q | No Attack | Pegasus | Parrot | DIPPER | GPT-3.5 |
> |-|-|-|-|-|-|
> | 8  | 94.2 | 43.4 | 54.2 | 22.0 | 20.4 |
> | 16 | 95.6 | 63.9 | 71.0 | 34.8 | 30.8 |
> | 32 | 96.4 | 72.4 | 80.0 | 46.0 | 42.2 |
> | 64 | 96.8 | 78.0 | 84.0 | 53.8 | 46.0 |
>
> - **Sentence Embedders**: Two alternative embedders achieved comparable, albeit slightly lower, performance. We retain all-mpnet-base-v2 as our default to ensure optimal semantic representations.
>
> | Embedder | No Attack | Pegasus | Parrot | DIPPER | GPT-3.5 |
> |-|-|-|-|-|-|
> | multi-qa-mpnet-base-dot-v1 | 97.0 | 68.2 | 80.6 | 39.3 | 38.0 |
> | all-distilroberta-v1 | 96.6 | 76.2 | 81.0 | 46.8 | 39.2 |
> | all-mpnet-base-v2 | 96.8 | 78.0 | 84.0 | 53.8 | 46.0 |
>
>
> ## Q3: Cross-domain Generalization
>
> We address the generalization concern with two new sets of experiments:
>
> - **New Domain**: We evaluated on the Natural Questions dataset (https://github.com/google-research-datasets/natural-questions). Despite the shift from sentence completion to question-answering, AliMark maintains outstanding performance.
>
> | Method | No Attack | Pegasus | Parrot | DIPPER | GPT-3.5 |
> |-|-|-|-|-|-|
> | SemStamp | 98.80 | 78.00 | 70.80 | 26.00 | 16.80 |
> | PMark | 97.20 | 87.00 | 90.60 | 40.20 | 26.00 |
> | AliMark | 99.00 | 91.20 | 92.80 | 59.00 | 56.60 |
>
> - **New Backbone**: Evaluations on gemma-3-1b-pt confirm AliMark continues to significantly outperform baselines.
>
> | Method | No Attack | Pegasus | Parrot | DIPPER | GPT-3.5 |
> |-|-|-|-|-|-|
> | SemStamp | 97.80 | 59.20 | 68.80 | 24.40 | 16.60 |
> | PMark | 93.20 | 51.80 | 64.40 | 21.80 | 15.80 |
> | AliMark | 98.20 | 71.80 | 80.20 | 47.60 | 45.00 |
>
> These results demonstrate AliMark's robustness across multiple domains and LLM architectures.
>
>
> We hope our responses have addressed your concerns. If you have any additional questions or suggestions, we would be more than willing to discuss them during this review period. Finally, if you feel that our responses satisfactorily address your concerns, we would be sincerely grateful for your kind consideration in raising the score for our submission. Thanks again for your valuable reviews.
>
> Sincerely,
>
> The Authors of Submission 587

---

> > ### Author Rebuttal · Reviewer_7gnv · 2026-04-03
> >
> > All of my concerns have been addressed.

---

> > > ### Author Response · Authors · 2026-04-08
> > >
> > > Dear Reviewer 7gnv,
> > >
> > > Thanks again for your positive feedback regarding our problem motivation, method design, and experimental results. We are glad to hear that our responses have fully addressed your concerns. We will ensure that all additional results and discussions from this rebuttal are incorporated into the final revised manuscript.
> > >
> > > Sincerely,
> > >
> > > The Authors of Submission 587

---

### Decision · Program_Chairs · 2026-04-30

**Decision:**

Accept (regular)

**Comment:**

The paper proposes a sentence-level watermarking framework that treats detection as a bit-sequence alignment problem to defend against perturbations from paraphrasers. The two weak-accept reviewers agree the method is "conceptually clean" and a "meaningful departure" from prior work. The two weak-reject reviewers raise concerns about novelty and evaluation scope, but these concerns are not well substantiated after the rebuttal. Reviewer YMka claims the method is closely related to existing approaches, yet does not cite any specific prior work achieving the same functionality. Reviewer hRnv states that the author's method is only slightly better than KGW, which is not supported by the numbers reported by the authors. Both reviewers acknowledge that the rebuttal partially addressed their concerns, but did not update their scores in response to the author's extensive experiments. I think the author's contributions could be interesting to a wider audience at ICML.